



# Stratospheric aerosol characteristics from space-borne observations: extinction coefficient and Ångström exponent

Elizaveta Malinina[1], Alexei Rozanov[1], Landon Rieger[2*], Adam Bourassa[2], Heinrich Bovensmann[1], John P. Burrows[1], and Doug Degenstein[2]

[1]Institute of Environmental Physics (IUP), University of Bremen, Bremen, Germany
[2]Institute of Space and Atmospheric Studies, University of Saskatchewan, Saskatoon, Canada
[*]now at Canadian Center for Climate Modeling and Analysis (CCCma), Environment Canada, Victoria (BC), Canada

*Correspondence to:* Elizaveta Malinina (malininaep@iup.physik.uni-bremen.de)

**Abstract.** Stratospheric aerosols are of a great importance to the scientific community, predominantly because of their role in climate, but also because accurate knowledge of aerosol characteristics is relevant for trace gases retrievals from remote sensing instruments. There are several data sets published which provide aerosol extinction coefficients in the stratosphere. However, for the instruments measuring in the limb viewing geometry, the use of this parameter is associated with uncertainties

resulting from the need to assume an aerosol particle size distribution (PSD) within the retrieval process. These uncertainties can be mitigated if PSD information is retrieved. While occultation instruments provide more accurate information on the aerosol extinction coefficient, in this study, it was shown that limb instruments have better potential for the PSD retrieval, especially during the background aerosol loading periods. A data set containing PSD information was recently retrieved from SCIAMACHY limb measurements and provides two parameters of the log-normal PSD for the SCIAMACHY operational

period (2002-2012). In this study, the data set is expanded by aerosol extinction coefficients and Ångström exponents calculated from the retrieved PSD parameters. Errors in the Ångström exponents and aerosol extinction coefficients are assessed using synthetic retrievals. For the extinction coefficient the resulting accuracy is within ±25%, and for the Ångström exponent, it is better than 10%. The recalculated SCIAMACHY aerosol extinction coefficients are compared to those from SAGE II. The differences between the instruments vary from 0 to 25% depending on the wavelength. Ångström exponent comparison with

SAGE II shows differences between 10% at 31 km and 40% at 18 km. Comparisons with SAGE II, however, suffer from the low amount of collocated profiles. Furthermore, the Ångström exponents obtained from the limb viewing instrument OSIRIS are used for the comparison. This comparison shows an average difference within 7%. The time series of these differences do not show signatures of any remarkable events. Besides, the temporal behavior of the Ångström exponent in the tropics is analyzed using the SCIAMACHY data set. It is shown, that there is no simple relation between the Ångström exponent and

the PSD because the same value of Ångström exponent can be obtained from an infinite number of combinations of the PSD parameters.





# 1  Introduction

According to the Fifth Assessment Report of IPCC (2013) (Intergovernmental Panel on Climate Change) clouds and atmospheric aerosols contribute the largest uncertainty to the estimates and interpretations of the Earth's changing energy budget. While there is a substantial number of publications and initiatives related to the role of the tropospheric aerosols (e.g. Popp et al., 2016), the role of stratospheric aerosols is currently not well addressed. Stratospheric aerosols influence climate through two major mechanisms. First, they scatter solar radiation and, during strong aerosol loading conditions, absorb the thermal infrared radiation upwelling from the troposphere; thus, changing the radiative budget of the Earth, and resulting in tropospheric cooling as well as stratospheric warming. As mentioned by Thomason and Peter (2006), the radiative effects of stratospheric aerosols are negligible during volcanically quiescent periods. However, after even small eruptions the influence of stratospheric aerosols on the climate becomes significant (Solomon et al., 2011; Fyfe et al., 2013). Furthermore, stratospheric aerosols play a key role in the stratospheric ozone depletion, which was reported to strengthen during the enhanced aerosol loading periods (Solomon, 1999; Ivy et al., 2017).

Accurate knowledge on the stratospheric aerosol loading is necessary for researchers in different fields. The atmospheric modelling community is particularly interested in this type of information, because climate models require knowledge about stratospheric aerosol to define the initial conditions and/or to assess the accuracy of their performance (Solomon et al., 2011; Fyfe et al., 2013; Brühl et al., 2015; Bingen et al., 2017). Other important applications of stratospheric aerosol data are the investigation of the effects of geoengineering (IPCC, 2013; Kremser et al., 2016) and use of stratospheric aerosol information to improve the retrieval of the stratospheric trace gases, e.g. water vapour (Rozanov et al., 2011) and ozone (Arosio et al., 2018; Zawada et al., 2018), from remote sensing instruments. Most commonly, stratospheric aerosols are characterized either by their extinction coefficient ($Ext$), or by one or several particle size distribution (PSD) parameters (e.g. median ($r_{med}$), effective ($r_{eff}$) or mode ($R_{mod}$) radius, distribution width parameter ($\sigma$), aerosol particle number density ($N$)). While for the instruments using the solar, lunar or stellar occultation measuring technique, $Ext$ retrieval is quite straightforward; for limb viewing instruments the retrieved $Ext$ is dependent on the assumed PSD parameters (more detailed in Sect. 3.2). Another complicating factor of $Ext$ is its wavelength dependency, which is also determined by the PSD. To retrieve all parameters defining a commonly assumed uni-modal log-normal PSD, at least three independent pieces of information at each altitude level are needed. However, this requirement is usually not satisfied for space-borne measurements and some assumptions have to be made (Thomason et al., 2008; Rault and Loughman, 2013; Rieger et al., 2014; Malinina et al., 2018). Some information about PSD can be obtained from the Ångström exponent (Ångström, 1929), which describes the wavelength dependency of $Ext$, although this parameter, if only one wavelength pair is used, can not be unambiguously transformed into the PSD parameters.

While there are long-term data sets of the aerosol PSD parameters from Optical Particle Counters (OPCs) (Deshler et al., 2003; Deshler, 2008), for the space borne remote sensing instruments they are much more limited. Known existing data sets were obtained from SAGE (Stratospheric Aerosol and Gas Experiment) II, an occultation instrument operated from 1984 to 2005 (Yue et al., 1989), (the data sets were described in Bingen et al. (2004); Thomason et al. (2008); Damadeo et al.



(2013)), as well as SAGE III on Meteor-3M platform (2001-2005) (Damadeo et al., 2013). In February 2017 the successor SAGE III mission on board of ISS (International Space Station) began its operation. However, the data product description and validation results have not been published by the time of writing. Another recent aerosol PSD data set including the $R_{mod}$ and $\sigma$ (distribution width parameter) was obtained from SCIAMACHY (Scanning Imaging Absorption Spectrometer

for Atmospheric CHartographY) limb data (Malinina et al., 2018). SCIAMACHY was one of the instruments operating on the Envisat satellite from 2002 till 2012 (Burrows et al., 1995; Bovensmann et al., 1999). More detailed information about the instrument can be found in Sect. 2.1. The data product v6.0 (Rieger et al., 2014) from OSIRIS (Optical Spectrograph and InfraRed Imager System) instrument on board on Odin satellite (Llewellyn et al., 2004) contains the Ångström exponent ($\alpha_{750/1530}$) from 2001 till 2012 (this instrument is described in Sect. 2.2 in more details). In addition, the theoretical basis for

the retrieval of PSD parameters and Ångström exponent was presented by Rault and Loughman (2013) for the OMPS (Ozone Mapping Profiler Suite) instrument, launched in 2011 (Jaross et al., 2014) and currently operational. However, no application to the real data were reported so far. For $Ext$ there are more existing data sets. Not only the above mentioned instruments have one or multiple $Ext$ products (e.g. there are multiple algorithms for SAGE II (Damadeo et al., 2013, and references therein), SCIAMACHY (Ovigneur et al., 2011; Taha et al., 2011; Ernst, 2013; Dörner, 2015; von Savigny et al., 2015; Rieger

et al., 2018b), OSIRIS (Bourassa et al., 2012; Rieger et al., 2018a) and OMPS (Loughman et al., 2018; Chen et al., 2018)), but other instruments employing limb or solar/lunar/stellar occultation measurement techniques also provide $Ext$ at different wavelengths. For example, GOMOS (Global Ozone Monitoring by Occultation of Stars), operated with SCIAMACHY on Envisat, provides $Ext$ at 525 nm (Vanhellemont et al., 2016). In addition, the space-based lidar CALIOP (Cloud-Aerosol Lidar with Orthogonal Polarization Lidar) provides measurements of the aerosol backscatter coefficient, which is then converted to

$Ext$. The $Ext$ profiles from CALIOP have the highest vertical resolution among the space borne instruments, but comparably sparse horizontal sampling (Vernier et al., 2011).

Since there are several continuous $Ext$ data sets which cover a wide time range, there are multiple comparison and merging possibilities for the evaluation of the long-term global behavior of stratospheric aerosols. SAGE II is considered to be one of the most reliable instruments in the era of occultation measurements, as it provided high-quality data for over 20 years, including

the period during and after the Mount Pinatubo eruption, the strongest volcanic eruption of the last decades. For that reason, this instrument is often used for merging and comparison activities (e.g. Thomason and Peter, 2006; Thomason, 2012; Ernst, 2013; Rieger et al., 2015; Kovilakam and Deshler, 2015; von Savigny et al., 2015; Kremser et al., 2016; Rieger et al., 2018b; Thomason et al., 2018).

In this manuscript we focus on the comparison of the Ångtröm exponents derived from SCIAMACHY, OSIRIS and to some

extent from SAGE II measurements. Furthermore, we present an evaluation of the errors in the $Ext$ and Ångström exponents, derived from SCIAMACHY PSD product. The manuscript has the following structure: Sect. 2 describes instruments and data used in the study, Sect. 3 presents an assessment of the limb and occultation instruments sensitivity to aerosol particles of different sizes. Sect. 4 includes the error assessment of the derived $Ext$ at different wavelengths, as well as the errors of the Ångström exponents calculated from the derived $Ext$. In Sect. 5 comparison of the Ångström exponents from SCIAMACHY,



OSIRIS and SAGE II is presented. The behavior of the Ångström exponents after the volcanic eruptions and the dependency of this parameter on the PSD parameters are discussed in Sect. 6.

## 2    Instruments and data

### 2.1    SCIAMACHY

SCIAMACHY was one of the instruments on the European Environmental satellite (Envisat), launched into the sun-synchronous orbit at 800 km altitude in March 2002 and operated till the loss of the contact in April 2012. SCIAMACHY made measurements in nadir, limb and solar/lunar occultation modes in 8 spectral channels, covering the spectral interval from 214 to 2386 nm with spectral resolution from 0.2 to 1.5 nm depending on the wavelength, and provided daily solar irradiance measurements. More detailed information can be found in Burrows et al. (1995); Bovensmann et al. (1999); Gottwald and Bovensmann (2011).

In this study we focus on the measurements performed in the limb viewing geometry. In this measurement mode the instrument scanned the atmosphere tangentially to the Earth's surface in the altitude range from about 3 km below the horizon, i.e., when the Earth's surface is still within the field of view of the instrument, up to about 100 km with a vertical step of 3.3 km and vertical resolution of 2.6 km. From SCIAMACHY limb observations $Ext$ at 750 nm (latest version is presented by Rieger et al. (2018b)) as well as two parameters of the aerosol PSD ($R_{mod}$ and $\sigma$) were retrieved (Malinina et al., 2018). The

PSD product was obtained by performing the retrieval with a fixed aerosol number density ($N$) profile taken from ECSTRA climatology (Fussen and Bingen, 1999), and using the limb radiances normalized to the solar irradiance and averaged over 7 wavelength intervals ($\lambda_1$=750±2 nm, $\lambda_2$=807±2 nm, $\lambda_3$=870±2 nm, $\lambda_4$=1090±2 nm, $\lambda_5$=1235±20 nm, $\lambda_6$=1300±6 nm, $\lambda_7$=1530±30 nm). The retrieval was performed from about 18 to 35 km. Spectral albedo was retrieved simultaneously with the PSD parameters, but only completely cloud free profiles were used in the retrieval. More detailed information about the

algorithm and the errors associated with a fixed $N$ profile can be found in Malinina et al. (2018).

### 2.2    OSIRIS

OSIRIS is a limb-viewing instrument on board the Swedish satellite Odin, having a sun-synchronous orbit at around 600 km altitude. The mission started its operation in February 2001 and continues working at the time of writing. OSIRIS consists of two instruments, an optical spectrograph (OS) and infrared imager (IRI). OS makes measurements of scattered solar light in

the spectral interval from 214 to 810 nm with 1 nm spectral resolution. Similarly to SCIAMACHY it observes the atmosphere tangentially to the Earth's surface in the altitude range from around 7 to 65 km with a vertical sampling of 2 km and vertical resolution of 1 km. IRI has a different measurement technique: it consists of three vertical photodiode arrays with 128 pixels each and filters 1260, 1270 and 1530 nm. Each pixel measures a line of sight at a particular altitude, and thus with each exposure the entire vertical profile covering around 100 km is created. Further information on the technical specifications of

OSIRIS is presented in Llewellyn et al. (2004).





Based on the measured information, $Ext$ profiles at 750 nm as well as Ångström exponent ($\alpha_{750/1530}$) were retrieved. To obtain $\alpha_{750/1530}$ the information at one wavelength of the optical spectrograph (750 nm) as well as measurements at 1530 nm from the infrared imager were used. As a reference spectrum the measurement at the higher tangent altitude was applied. In the retrieval, $r_{med}$ and $N$ were fitted assuming fixed $\sigma$=1.6, and the obtained values were used to calculate the Ångström exponent

and $Ext$ at 750 nm. Due to a lack of the absolute calibration for the infrared imager, only albedo at 750 nm was retrieved. A detailed description of the algorithm and the products is given by Rieger et al. (2014).

## 2.3 SAGE II

SAGE II was a solar occultation instrument on the Earth Radiation Budget Satellite (ERBS). It operated from October 1984 to August 2005 in an orbit with 57° inclination at about 600 km altitude (Barkstrom and Smith, 1986). SAGE II was a Sun

photometer with seven silicon photodiods with filters at 386, 448, 525, 600, 935 and 1020 nm wavelengths. During each sunrise and sunset encountered by the satellite the instrument measured solar radiance attenuated by the Earth's atmosphere. The measurements were provided from the cloud top to about 60 km with the vertical resolution of about 0.5 km. However, the spacial coverage of the measurements is quite sparse, as there is one sunrise and one sunset event per orbit. This results in 30 profiles per day, unlike SCIAMACHY and OSIRIS, which provide about 1400 profiles per day each. More technical

information on SAGE II can be found in McCormick (1987).

For this study we used v7.0 of the SAGE II product, which is described in detail by Damadeo et al. (2013). In this version $Ext$ profiles at 1020, 525, 452, and 386 nm are provided. For their retrieval, first, the slant-path transmission profiles were calculated at each wavelength, then using the spectroscopy data slant-path optical depth profiles were obtained for each of the retrieved species. With an "onion-peeling" technique the optical depth profiles were inverted to obtain $Ext$ profiles. Later,

based on the $Ext$ at 525 nm and 1020 nm, $r_{eff}$ as well as surface area density were obtained (Thomason et al., 2008).

## 3 Sensitivity of measurements to aerosol parameters

### 3.1 Aerosol parametrisation

Stratospheric aerosols are commonly represented by spherical droplets containing 75% $H_2SO_4$ and 25% of $H_2O$ with particle sizes distributed log-normally (Thomason and Peter, 2006). Though in some studies using in-situ instruments bimodal PSD is

employed (e.g. Deshler et al., 2003; Deshler, 2008), for the space borne remote sensing instruments a unimodal distribution is most commonly considered (Damadeo et al., 2013; Rieger et al., 2014; von Savigny et al., 2015; Malinina et al., 2018):

$$\frac{dn}{dr} = \frac{N}{\sqrt{2\pi}\ln(\sigma)r} \exp\left(-\frac{(\ln(r_{med}) - \ln(r))^2}{2\ln^2(\sigma)}\right),$$
(1)

where $N$ is the aerosol particle number density, $r_{med}$ is the median radius and $\ln(\sigma)$ is the standard deviation of the $\frac{dn}{d\ln(r)}$ function. In some studies mode radius is used instead of the median radius ($r_{med}$) for the aerosol parametrisation. The former

is defined as $R_{mod} = r_{med}/\exp(\ln^2(\sigma))$. In addition, Malinina et al. (2018) used the standard deviation of $dn/dr$ function,





which is referred to as the absolute distribution width:

$$w = \sqrt{r_{med}^2 \exp(\ln^2(\sigma))(\exp(\ln^2(\sigma)) - 1)}, \tag{2}$$

because this parameter is easier for visual interpretation than $\sigma$, which is most commonly used in the aerosol parametrizations. In this study, similarly to Malinina et al. (2018), $\sigma$ will be used when describing the retrieval settings, while $w$ will be used in the results discussion.

As mentioned in Sect. 1, PSD parameters uniquely describe a log-normal distribution of the aerosol particle sizes, although often due to a lack of the information $Ext$ is retrieved. Aerosol extinction coefficient at the wavelength $\lambda$ is defined as

$$Ext_\lambda = \beta_{aer}(r_{med}, \sigma, \lambda...)N, \tag{3}$$

where $\beta_{aer}$ is the aerosol extinction cross section, which is dependent on the aerosol PSD (e.g. Liou, 2002). Some limited information about PSD is given by the Ångström coefficient or Ångström exponent, $\alpha$, which was used in the empirical relation introduced by Ångström (1929):

$$\frac{Ext_{\lambda_1}}{Ext_{\lambda_2}} = \left(\frac{\lambda_1}{\lambda_2}\right)^{-\alpha}. \tag{4}$$

However, the usage of $\alpha$ is associated with certain issues. In his work Ångström (1929) noted that the diameter of the particles calculated from $\alpha$ shows only an approximate coincidence with the average aerosol diameter directly measured. Furthermore he states, that the changes in the size of the particles do not necessarily lead to the changes in $\alpha$. Another complication is related to the fact, that $\alpha$ value is spectral dependent and thus changes based on the wavelength pair used for its calculation (e.g. Rieger et al., 2014). It is important to recall, that for limb instruments the retrieved values of $Ext$ depend on the assumed PSD parameters. More detailed information about the influence of the errors in PSD parameters on the retrieved $Ext$ can be found in Rieger et al. (2018b); Loughman et al. (2018).

## 3.2  Measurement sensitivity

As mentioned before, occultation and limb scatter instruments employ different measurement approaches, resulting in different sensitivity to the aerosol parameters. When discussing occultation measurements in this study, we assume the measurements by solar occultation instruments. Such instruments register the solar radiation transmitted through the atmosphere during sunrise and sunset events as seen from the satellite. In contrast, limb scatter instruments measure profiles of the solar radiation scattered by the atmosphere. As it is presented in e.g. Rozanov et al. (2001), the radiative transfer equation is solved for occultation and limb observations in very different ways. The direct radiance as registered by an occultation instrument is described by:

$$I_{dir}(\Omega) = I_0 \exp\left(-\int_0^s k(\hat{s})d\hat{s}\right), \tag{5}$$

where $I_0$ is the incident solar flux, $\Omega$ is an angle, defining the radiance propagation direction, $s$ is the full path length through the atmosphere along the solar beam, and $k$ is the extinction coefficient.





The diffuse radiance, which is observed by a limb scatter instrument is given by:

$$I_{dif}(\Omega) = \int_0^{s_{LOS}} \frac{1}{4\pi} \bigg( \int_\Omega \big( \eta_R p_R(\Omega,\Omega') + \eta_a p_a(\Omega,\Omega') \big) I_{dif}(\Omega') d\Omega' +$$

$$+ \big( \eta_R p_R(\Omega,\Omega_0) + \eta_a p_a(\Omega,\Omega_0) \big) I_0 \exp\big( - \int_0^{s_{Sun}} k(\hat{s}) d\hat{s} \big) \bigg) e^{-\tau(s)} ds. \tag{6}$$

Here, $s_{LOS}$ stands for the full path along the line of sight of the instrument, $\eta_R$ and $p_R$ for the Rayleigh scattering coefficient and the phase function, $\eta_a$ and $p_a$ for the scattering coefficient and the phase function of the aerosol, $\Omega_0$ for the solar beam propagation direction, $s_{Sun}$ for the full path along the solar beam and $\tau$ for the optical depth along the light-of-sight.

Comparing Eqs. (5) and (6) the one can see, that it is quite straightforward to derive $k$ from $I_{dir}$. In the wavelength intervals without any other absorber features $k$ represents a sum of the aerosol extinction coefficient and Rayleigh scattering coefficient, i.e. $Ext + \eta_R$. In contrast, to obtain $I_{dif}$ using Eq. 6 an iterative approach is needed. Furthermore, $I_{dif}$ depends on the product of $p_a$ and $\eta_a$. In turn, both, $p_a$ and $\eta_a$ are determined by the aerosol PSD parameters. Thus, in most of the $Ext$ retrieval algorithms which rely on limb measurements an assumption on the PSD parameters is used, and those are kept fixed during the retrieval process. In addition, $p_a$ is a function of the scattering angle. The issue related to the dependency of the limb radiances on the solar scattering angle ($SSA$) is well known, and was discussed by Ernst (2013); Rieger et al. (2014, 2018b); Loughman et al. (2018).

To understand how the differences in the measurement techniques influence the instrument sensitivity to aerosols, extended analysis is provided below. In some previous studies (e.g Twomey, 1977; Thomason and Poole, 1993; Rieger et al., 2014) the analysis of so-called kernels was used to show the contribution of the particles of different sizes to the observed radiance. According to Twomey (1977) the measured intensity of the scattered light can be presented as:

$$I(\lambda) = \int_0^\infty K(\lambda, r) n(r) dr, \tag{7}$$

where $r$ is a radius of the particle and $K$ is a kernel. For the measurements of the transmitted light the following equation is appropriate (Twomey, 1977):

$$\ln(I(\lambda)/I_0(\lambda)) = \int_0^\infty K(\lambda, r) n(r) dr. \tag{8}$$

Whereas for the measurements of scattered solar light $K$ does not have an analytic representation, for the occultation measurements it is given by $r^2 Q_e(r/\lambda)$, where $Q_e$ is the Mie extinction efficiency. Besides, the right sides of the Eqs. (7) and (8) have the same form, although they refer to different left sides. Indeed, for the scattered light measurements the left side is represented by $I(\lambda)$, while for the transmission the left side is $\ln(I(\lambda)/I_0(\lambda))$, which according to the Eq. (5) is $\tau = -\int_0^s k(\hat{s}) d\hat{s}$. Thomason and Poole (1993) derived $K$ for the extinction measurements and showed it for SAGE II. In their research $K$ had units of $m^{-1}$, since they were assessing it per unit volume of air. For the limb measurements, $K$ was derived for the single-scatter





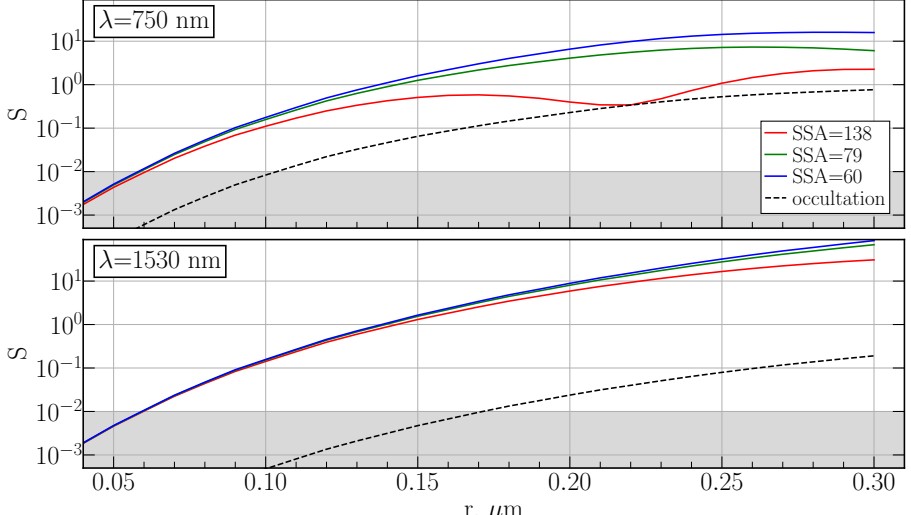

**Figure 1.** Modelled sensitivity, $S$, at 21.7 km for range of particles radii ($r$). The simulations were performed for different limb geometries with different solar scattering angles, $SSA$, and for one occultation geometry.

radiance by Rieger et al. (2014), and the resulting $K$ was dimensionless. It should be also noted, that for their study Rieger et al. (2014) preferred to calculate $K$ for OSIRIS numerically, stating, that the derived formula contains too many approximations. Thus, $K$ can be used in the assessment of the sensitivity of the instruments employing the same measurement technique, but is not suitable for the interinstrumental comparisons. As one objective of our study is the comparison of the sensitivities of the limb and occultation measurements to the particles of the different sizes, we will not follow the approach of Twomey (1977). Instead, we define the dimensionless measurement sensitivity, $S$, to the aerosol particles of a certain size as a change in the intensity of the observed radiation with respect to the aerosol free conditions. Thus, $S$ is defined by:

$$S(\lambda, r_i) = \frac{I(\lambda) - I_R(\lambda)}{I_R(\lambda)}, \qquad (9)$$

where $I(\lambda)$ is the radiance including both, Rayleigh and aerosol signals, and $I_R$ is the Rayleigh signal. When $S=0$, the radiance has no contribution from the aerosol extinction or scattering. With increasing $S$ increases the aerosol contribution to the measured radiance.

The quantitative assessment of $S$ is made by modelling the intensities with the radiative transfer model SCIATRAN Rozanov et al. (2014) for limb measurements (on the example of SCIAMACHY limb geometries) and for occultation measurements. The intensities were modelled for the distributions with $R_{mod}$ varying from 0.04 to 0.30 $\mu$m with the step ($\Delta r$) of 0.01 $\mu$m. For each distribution, $\sigma$ has been chosen such that $w$ is equal to 0.01 $\mu$m (this corresponds to the chosen $\Delta r$). $N$ was considered to be the same for all simulations.

For the above described simulations $S$ at $\lambda$=750 nm and $\lambda$=1530 nm are presented in Fig.1 for occultation (dashed black line) and for limb measurements (colored solid lines). As limb radiances depend on $SSA$, the simulations were done for three



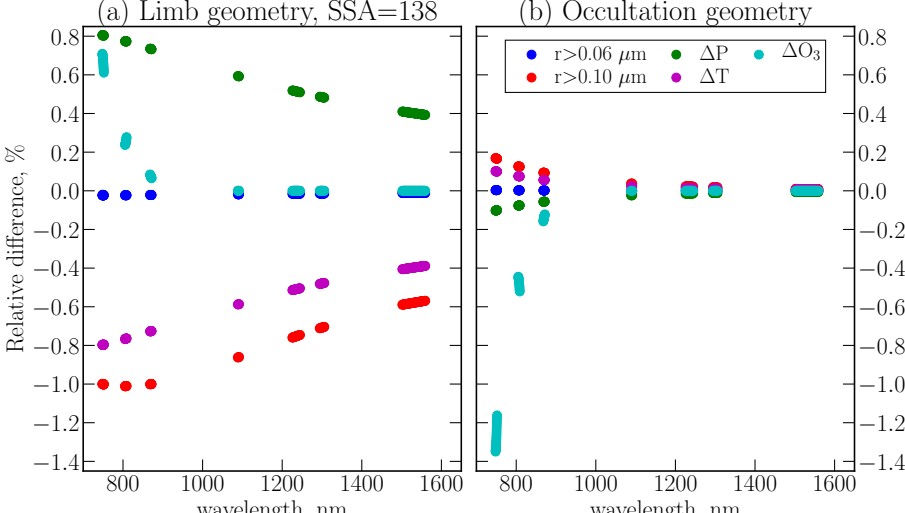

**Figure 2.** Relative changes in the radiance at 21.7 km for limb (panel (a)) and occultation (panel (b)) geometry. The responses in radiance due to 10% changes in temperature, atmospheric pressure and ozone concentration are presented by magenta, green and cyan dots, respectively. The cut off in PSD at 0.06 $\mu$m is depicted with blue dots, and the cut off at 0.10 $\mu$m is presented with red dots.

different observational geometries: $SSA$=60° (blue line), $SSA$=79° (green line) and for $SSA$=138° (red line). The angles from 60 to 140° represent the $SSA$ range for SCIAMACHY measurements in the tropical region. The grey shaded area shows $S$<0.01. We believe, that this empirical value stays for a typical uncertainty of the measurement-retrieval system, caused by the uncertainties in the radiative transfer modelling. A justification for this is provided further in this section.

As it is seen from Fig. 1, $S$ for limb radiances at both wavelengths are obviously higher than those for the occultation measurements. Moreover, for all the $SSA$s the limb curves enter the shaded grey area at around 0.06 $\mu$m, while the occultation curves enter this area at about 0.11 $\mu$m at $\lambda$=750 nm and at 0.20 $\mu$m at $\lambda$=1530 nm. These results agree with those presented by Thomason and Poole (1993) for SAGE II measurements, and illustrate the well-known statement that the occultation measurements are insensitive to the particles with the radii smaller than 0.1 $\mu$m (see e.g. Thomason et al., 2008; Kremser et al.,

2016). Better sensitivity to the smaller particles observed for the limb measurements is explained by the fact that the aerosol PSD contributes in very different ways into the observed radiances for limb and occultation measurements. While $I_{dir}$ depends inverse exponentially on $k$ (or $Ext$), $I_{dif}$ is, to a first order, proportional to the product of $p$ and $\eta$. Both, $p$ and $\eta$, depend on the PSD parameters, and as a result, limb radiances tend to be more sensitive to aerosol particles of the smaller size, and thus provide more accurate PSD parameters during the the "background" aerosol loading conditions.

To justify the choice of the sensitivity threshold, we provide an example of the relative differences in the the modelled radiance due to changes in different factors for both, limb ($SSA$=138°) and occultation, geometries assuming randomly picked PSD with $R_{mod}$=0.08 $\mu$m and $\sigma$=1.6. These relative differences at 21.7 km for different wavelengths are presented in Fig. 2, with the results for limb geometry depicted in panel (a) and for occultation geometry in panel (b). For each simulation the whole




profile of one of the following parameters was increased by 10%: temperature (magenta dots), atmospheric pressure (green dots) and ozone concentration (cyan dots). For simplicity reasons, we choose to perturb the whole profile, rather than values at the particular altitudes. Changes in ozone concentration by 10% are reasonable as they reflect the remaining uncertainties in the ozone profiles retrieved from the space-borne measurements across the relevant altitude range (Tegtmeier et al., 2013).

Additionally, we depicted with the blue dots the changes in the radiances by assuming the concentration of the aerosol particles with $r <= 0.06$ $\mu$m being equal to zero (cut off at 0.06 $\mu$m) and with the red dots assuming the cut off at 0.10 $\mu$m (particles with $r <= 0.10$ $\mu$m were not considered). Figure 2 shows, that for the limb geometry the relative changes in the radiance due to changes in temperature, pressure or ozone concentration are within 0.8% in the considered wavelength interval. For the PSD cutoff at 0.06 $\mu$m the changes are about 0.03%, however, for the cut off at 0.10 $\mu$m the changes are slightly larger than

1% for the wavelength interval shorter than 1000 nm, and and reducing to 0.6% at the longer wavelengths. For the occultation geometry the changes are somewhat different. Namely, the changes in pressure or temperature are comparable with the changes in the PSD cut off, but the changes in ozone concentration contribute up to 1.4% in the radiance at 750 nm. Such behaviour of relative changes in the intensities shows that in the considered wavelength interval 1% provides a realistic estimation of the uncertainties from the radiative transfer modelling for both geometries, even though this uncertainty is caused by different

factors. For limb geometry, the changes due to PSD cut off at $r=0.06$ $\mu$m are within this threshold, however, with the cut off at $r=0.10$ $\mu$m the relative difference in the radiances exceeds it, and thus this change is considered as detectable. For the occultation measurements, both PSD cut offs result in smaller changes than those coming from the radiative transfer modelling uncertainty.

  Summarizing Sect. 3.2, it can be concluded, that due to differences in the underlying radiative transfer processes, limb and

occultation instruments have intrinsically different sensitivity to stratospheric aerosol parameters. Limb radiances are more sensitive to the smaller aerosol particles, which is expected to result in a more accurate PSD parameters retrieval. This is particularly the case during the background aerosol loading periods when smaller particles prevail. However, $Ext$ retrieval from limb instruments suffers from the uncertainties due to assumed PSD and the $SSA$ dependency. On the contrary, the retrieval of $Ext$ from the occultation radiances is more straightforward, but the threshold of the sensitivity of the occultation

measurements to the aerosol particle sizes is somewhat lower in comparison to the limb instruments.

## 4 Error assessment

### 4.1 Extinction coefficients errors

As discussed above, aerosol extinction coefficient data bases are widely used in the stratospheric aerosol research. In order to make SCIAMACHY aerosol PSD product comparable with the products from other satellite instruments, $Ext$ at four wave-

lengths were calculated and then used to derive the Ångström exponents at two wavelength pairs.

  As it was mentioned in Sect. 2.1, SCIAMACHY PSD product provides $R_{mod}$ and $\sigma$. These parameters were retrieved assuming a fixed $N$ profile. Thus, during the volcanically active periods, an inadequate assumption of $N$ results in errors in the retrieved $R_{mod}$ and $\sigma$. To assess how this assumption affects resulting $Ext$ and Ångström exponent, synthetic retrievals were



performed. The limb radiances were simulated with the known "true" parameter settings and then used in the retrieval instead of the measurement spectra. Applying this approach, Malinina et al. (2018) showed, that $R_{mod}$ was retrieved with an accuracy of about 20% even if the true value of $N$ was a factor of 2 higher than the value assumed in the retrieval. For scenarios with the perturbed $N$ profile, $\sigma$ is retrieved with about 10% accuracy, while an accuracy of about 5% is reached for the profiles with

unperturbed $N$.

To extend the error assessment of SCIAMACHY PSD product, we use the same scenarios and the same modelled radiances as (Malinina et al., 2018) to evaluate errors in $Ext$ resulting from the errors in the retrieved $R_{mod}$ and $\sigma$. For this study $Ext$ was calculated employing Mie theory and Eq. (3) at 525, 750, 1020 and 1530 nm using the retrieved PSD information and the $N$ profile assumed in the retrieval (exponentially decreasing from 15.2 cm$^{-3}$ at 18 km to 0.5 cm$^{-3}$ at 35 km). In addition,

$Ext_{750}$ was retrieved from the simulated radiances using an algorithm similar to SCIAMACHY v1.4 (Rieger et al., 2018b), but with the normalization to the solar spectrum and using the phase function which was calculated from the retrieved $R_{mod}$ and $\sigma$ at each altitude. For this retrieval the albedo value was set to the value resulting from the PSD parameters retrieval. To distinguish the two approaches to calculate extinctions we denominate $Ext$, obtained with Eq. (3) as "calculated" and $Ext$, retrieved using the corrected PSD as "retrieved".

Five scenarios for a typical observational geometry in the tropical region were used. Four of them: "small", "background", "unperturbed" and "volcanic" were simulated with the same $N$ profile as the one used for the retrieval and the aerosol PSD parameters listed in Tab. 1. For the scenario "volcanic (2N)" $N$ profile was multiplied by the factor of 2 below 23 km (as discussed in Malinina et al. (2018) this approach is considered to be realistic for the SCIAMACHY operation period). The scenarios are summarized in Tab. 1. For all scenarios the surface albedo was set to 0.15 at all wavelengths (perturbed by 0.35 in

comparison to the first guess value 0.5). The measurement noise was simulated by adding the Gaussian noise to the simulated radiances. The signal-to-noise ratios were assessed from the SCIAMACHY measurements. The retrievals were done using 100 independent noise sequences to ensure reliable statistics.

Panel (a) of Fig. 3 shows the median calculated $Ext_{750}$ profiles with solid lines and median retrieved $Ext_{750}$ profiles with dashed lines. The "true" values are shown by the dotted lines. The colors corresponding to each scenario are listed in

Tab. 1. Panel (b) shows the median relative errors for both calculated and retrieved $Ext_{750}$. Solid shaded areas show $\pm 1$ standard deviation for the calculated profiles, while the striped ones denote $\pm 1$ standard deviation for the retrieved profiles. The maximum relative errors for the calculated aerosol extinction coefficients at other wavelengths is presented in Tab 1. As the altitudinal behaviour of the extinction coefficients at the other wavelengths is the same as for the $Ext_{750}$ profiles, we show the results only for one wavelength.

As it follows from Tab. 1, the relative errors for the scenarios with unperturbed $N$ do not exceed 20%. As expected, for the "volcanic (2N)" scenario, the errors are slightly higher and vary depending on the wavelength from 19% to 31%. For all scenarios the largest errors are observed for $Ext_{525}$. This is most likely because for the retrieval of PSD parameters only the wavelengths longer than 750 nm were taken into consideration, while the information from the visible and UV parts of the spectrum has no contribution to the retrieval.



**Table 1.** Selected scenarios and associated maximum relative errors in the calculated extinction coefficients.

| Name | True | | Color[*] | Perturbation | Max. | | | |
|---|---|---|---|---|---|---|---|---|
| | $R_{mod}$ | $\sigma$ | | | $\epsilon_{Ext_{525}}$ | $\epsilon_{Ext_{750}}$ | $\epsilon_{Ext_{1020}}$ | $\epsilon_{Ext_{1530}}$ |
| Small | 0.06 $\mu$m | 1.7 | cyan | $R_{mod}, \sigma$ | 20% | 17% | 15% | 12% |
| Background | 0.08 $\mu$m | 1.6 | blue | $R_{mod}, \sigma$ | 10% | 9% | 8% | 7% |
| Unperturbed | 0.11 $\mu$m | 1.37 | brown | unpert. | 4% | 4% | 4% | 4% |
| Volcanic | 0.20 $\mu$m | 1.2 | green | $R_{mod}, \sigma$ | 7% | 9% | 10% | 11% |
| Volcanic (2N) | 0.20 $\mu$m | 1.2 | red | $R_{mod}, \sigma, N$ | 31% | 26% | 22% | 19% |

[*] Color of the lines in Fig. 3.

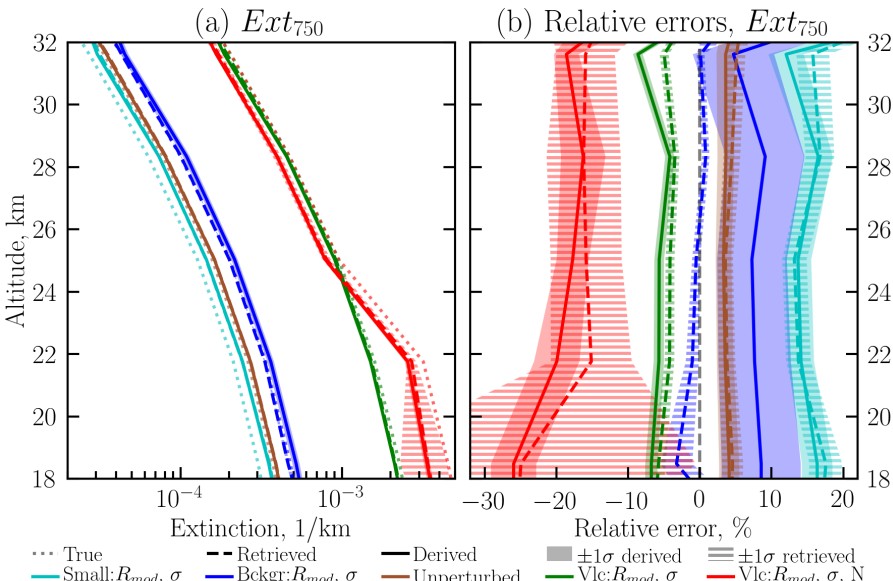

**Figure 3.** Profiles of $Ext_{750}$ (a) and their relative errors (b) for a typical tropical observation geometry. The solid lines show the calculated from PSD product profiles, dashed lines depict the directly retrieved profiles, while the dotted lines represent the true values. The shaded areas stand for $\pm$ 1 standard deviation. The scenarios used for the simulations are listed in Table 1.

    Analyzing Fig. 3 it is important to mention that retrieved $Ext_{750}$ barely differs from that calculated. For "small", "unperturbed" and "volcanic" scenarios the solid and dashed lines are very close to each other, and the median relative errors of the retrieved profiles lay mostly inside the standard deviation of calculated profiles. For the "background" conditions the retrieved and the calculated profiles have the same shape, but the retrieved profiles are about 8% more accurate. As it was suggested that

5   the difference between calculated and retrieved profiles of $Ext$ might possibly be used to correct $N$ for the PSD retrieval, it is most important to analyze the $Ext$ profiles for the scenario with the perturbed $N$ profile ("volcanic (2N)"). As it can be seen from the Fig. 3 (a) the retrieved $Ext_{750}$ shows similar altitudunal behaviour as the calculated profile, although the retrieved





profile has larger standard deviation at the lowermost altitude. Similarly to the calculated profile, the retrieved profile is about 25-30% lower than the "true" one. This leads to the conclusion, that an additional retrieval of $Ext$ with the corrected PSD or fixing $Ext$ during the retrieval does not provide any additional information about the aerosol PSD, and some other independent data or constraint is needed to retrieve all three parameters. One possible way would be to combine limb and occultation measurements, using the later to constraint $N$. Another possibility to constraint $N$ would be through the use of collocated profiles of stratospheric $H_2SO_4$ concentrations. For SCIAMACHY the use of the MIPAS (Michelson Interferometer for Passive Atmospheric Sounding) $H_2SO_4$ volume mixing ratio data set (Günther et al., 2018) might be appropriate. However, synergistic use of the data from two different instruments is not straightforward, and is a subject of the further studies.

## 4.2 Ångström exponents errors

Combining Eqs. (3) and (4), the Ångström exponent can be obtained as

$$\alpha_{\lambda_1/\lambda_2} = -\frac{\ln(\beta_{aer}(\lambda_1, r_{med}, \sigma)/\beta_{aer}(\lambda_2, r_{med}, \sigma))}{\ln(\lambda_1/\lambda_2)}. \tag{10}$$

Equation (10) shows, that Ångström exponent is not directly dependent on $N$, thus only the errors from $R_{mod}$ and $\sigma$ influence the derived $\alpha_{\lambda_1/\lambda_2}$. To assess this influence, the Ångström exponents were calculated using Eq. (10) and the calculated $Ext$ from the synthetic retrievals discussed in Sect. 4.1. While from SCIAMACHY PSD product the Ångström exponents can be calculated at any wavelength pair, SAGE II and OSIRIS provide only $\alpha_{525/1020}$ and $\alpha_{750/1530}$ respectively. Thus, we limit our analysis to those values. The "true" Ångström exponent values, the scenario summaries (described in the previous section) as well as the maximum absolute and relative errors for $\alpha_{525/1020}$ and $\alpha_{750/1530}$ are presented in Tab. 2. As for $Ext$ analysis, we show the altitudinal behavior only for one wavelength pair ($\alpha_{750/1530}$), while the results for the other pair are very similar. In panel (a) of Fig. 4 the median derived Ångström exponents are presented with solid lines, and the "true" values are shown by dashed lines; in the panel (b) median relative errors for the chosen scenarios are depicted. For both panels shaded areas show $\pm 1$ standard deviation.

Analysis of Tab. 2 and Fig. 4 leads to the conclusion that the relative error in the Ångström exponent for all scenarios is below 10% for $\alpha_{525/1020}$, and less than 5% for $\alpha_{750/1530}$. As expected, the largest errors are seen for the "volcanic (2N)" scenario, where $N$ profile was perturbed and the errors in $R_{mod}$, $\sigma$ and $Ext$ were the largest. For all scenarios the largest errors are observed at the lowermost retrieved altitude, e.g. for $\alpha_{750/1530}$ the errors above 21.3 km do not exceed 2.5%. The absolute error for $\alpha_{525/1020}$ is less than 0.12 for the scenarios with unperturbed $N$ and about 0.2 for the scenario with $N$ perturbed by a factor of 2. For $\alpha_{750/1530}$ the absolute errors are even smaller, in particular, for the scenarios with the same $N$ as used for the retrieval the errors are smaller than 0.1, and for the "volcanic (2N)" scenario the difference between the true and derived Ångström exponent is 0.15.

Summarizing Sect. 4 it can be concluded, that errors in the calculated $Ext$ for the background scenarios do not exceed 20%, and are about 20-25% for the cases with the perturbed $N$. The largest errors are observed for the aerosol extinction coefficient at 525 nm, most likely due to the missing information from the visible spectral range in the PSD retrieval. The retrieval results for $Ext$ with the retrieved PSD parameters barely differs from the aerosol extinction coefficients calculated from PSD product,



**Table 2.** Selected scenarios and associated maximum absolute (relative) errors in Ångström exponents.

| Name | True | | | | Color[*] | Perturbation | Max. | |
|---|---|---|---|---|---|---|---|---|
| | $R_{mod}$ | $\sigma$ | $\alpha_{525/1020}$ | $\alpha_{750/1530}$ | | | $\epsilon_{\alpha_{525/1020}}$ | $\epsilon_{\alpha_{750/1530}}$ |
| Small | 0.06 $\mu m$ | 1.7 | 2.18 | 2.76 | cyan | $R_{mod}, \sigma$ | 0.12 (4.8%) | 0.10 (4.3%) |
| Background | 0.08 $\mu m$ | 1.6 | 2.22 | 2.84 | blue | $R_{mod}, \sigma$ | 0.05 (2.4%) | 0.06 (2.1%) |
| Unperturbed | 0.11 $\mu m$ | 1.37 | 2.76 | 3.36 | brown | unpert. | 0.01 (0.3%) | 0.01 (0.2%) |
| Volcanic | 0.20 $\mu m$ | 1.2 | 2.41 | 3.12 | green | $R_{mod}, \sigma$ | 0.04 (1.6%) | 0.03 (1.0%) |
| Volcanic (2N) | 0.20 $\mu m$ | 1.2 | 2.41 | 3.12 | red | $R_{mod}, \sigma, N$ | 0.20 (8.3%) | 0.15 (4.7%) |

[*] Color of the lines in Fig. 4.

In the last 2 columns maximum absolute error for the profile is given by the number without brackets, while the maximum relative error is presented in brackets.

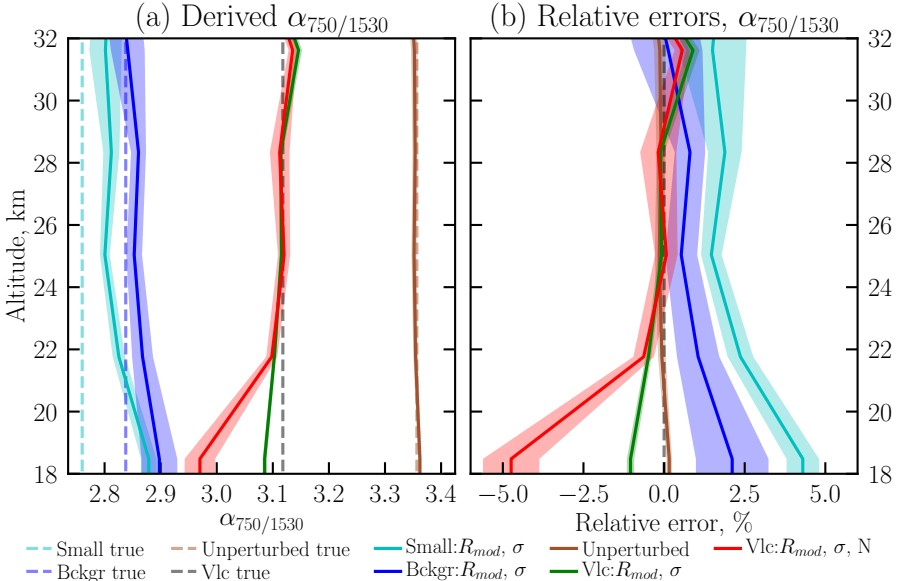

**Figure 4.** Ångström exponent profiles ($\alpha_{750/1530}$) (a) and their relative errors (b) for a typical tropical observation geometry. The solid lines show the derived from PSD product profiles, while the dashed lines represent the true values. The shaded areas stand for $\pm 1$ standard deviation. The scenarios used for the simulations are listed in Table 2.

and thus, cannot be used to improve the knowledge of $N$. For the Ångström exponent the errors are much smaller, as they cancel out in the ratio of the aerosol extinction coefficients and depend only on the aerosol PSD. For $\alpha_{525/1020}$ the relative error does not exceed 10%, and for $\alpha_{750/1530}$ the relative error is below 5%. For both Ångström exponents the absolute errors are less than 0.2 for the cases with the perturbed $N$ profile and are even less than 0.1 for unperturbed $N$.



## 5 Comparison of the measurement results

### 5.1 Extinction coefficients comparison with SAGE II

As discussed in the previous sections, SAGE II was an outstanding instrument, providing aerosol extinction coefficients, and there have been several comparisons performed using its data. For the SCIAMACHY PSD product the comparisons with SAGE II are associated with some challenges. First, SCIAMACHY and SAGE II have only a 3 year period of overlap. Second, SAGE II was an occultation instrument, providing about 30 vertical profiles per day, which resulted in 57 collocated profiles with SCIAMACHY (collocation criteria are $\pm 5°$ latitude, $\pm 20°$ longitude and $\pm 24$ hours) for the entire overlap period because the SCIAMACHY PSD retrieval is currently limited to completely cloud free profiles in the tropical zone (20°S-20°N). This amount of collocated profiles is not enough for an indepth investigation, however a rough assessment of the consistency of the data from both instruments can be provided. Another issue related to the SCIAMACHY and SAGE II comparison is the difference in the measurement techniques and thus a different sensitivity to aerosol properties. As was discussed in Sect. 3.2, SAGE II as an occultation instrument is less sensitive to the smaller particles, although $Ext$ retrieval from SAGE II is associated with smaller uncertainties. The limb measurements from SCIAMACHY, in turn, are more sensitive to the particles with $r <0.10$ $\mu$m, although the direct retrieval of $Ext$ is associated with some issues (see Sect. 3.2). The comparison of the $r_{eff}$ from SAGE II and SCIAMACHY, presented in Malinina et al. (2018) is expected to be influenced by these differences, because the instruments overlap period is considered to be volcanically quiescent with a prevailing amount of smaller particles. Here, we present the comparison of $Ext$ retrieved from SAGE II and that calculated from SCIAMACHY PSD product, which is expected to be more reliable than direct comparison of $r_{eff}$.

To perform the comparison, SCIAMACHY aerosol extinction coefficients at 525, 750 and 1020 nm were calculated with Eq. (3), considering the same $N$ profile as used in the PSD retrieval. As SAGE II did not have a 750 nm channel, $Ext_{750}$ for this instrument was calculated with Eq. (4) from $Ext_{525}$ and $Ext_{1020}$ using $\alpha_{525/1020}$. To assess a possible uncertainty associated with the usage of the Ångström exponent when calculating $Ext_{750}$ from SAGE II data, SCIAMACHY $Ext_{750}$ was additionally calculated using the same approach. To distinguish between two different methods of $Ext_{750}$ calculation, we use $Ext_{750}(PSD)$ for the one derived from PSD product with Eq. (3), and $Ext_{750}(\alpha_{525/1020})$ for that calculated using the Ångström exponent. As SCIAMACHY and SAGE II have different vertical resolution, SAGE II data was smoothed to the coarser SCIAMACHY vertical resolution and then interpolated onto the SCIAMACHY vertical grid. The mean relative differences between SCIAMACHY and SAGE II for $Ext_{525}$ (blue line), $Ext_{1020}$ (green line), $Ext_{750}(PSD)$ (red line) and $Ext_{750}(\alpha_{525/1020})$ are presented in Fig. 5. The shaded areas show the standard error of the mean. We prefer to depict the standard error of the mean instead of the standard deviation to make the figures less busy.

As it can be seen from Fig. 5, for all wavelengths the errors for the derived extinction coefficients are below $\pm 25\%$, which is within the reported precision of the extinction coefficients for SCIAMACHY. For $Ext_{1020}$ the shape of the relative difference follows the one reported earlier in Malinina et al. (2018) for the differences between SCIAMACHY and SAGE II effective radii, but with slightly different values (-20 to 10% for $Ext_{1020}$ versus -30 to 0% for $r_{eff}$). Such behaviour is expected, because $r_{eff}$ from SAGE II is obtained using the $Ext_{1020}$ and $Ext_{525}$ (see Sect. 2.3). The differences in the $Ext_{525}$, $Ext_{750}(PSD)$ and



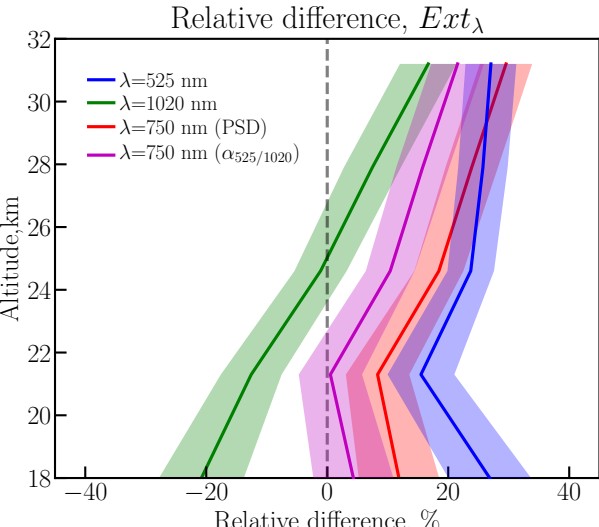

**Figure 5.** Mean relative difference ( $200 \times$(SCIAMACHY-SAGE II)/(SCIAMACHY+SAGE II)) between extinction coefficients at 525 nm (blue line), 1020 nm (green line), 750 nm obtained directly from PSD (red line) and 750 nm converted with $\alpha_{525/1020}$ (magenta line) from collocated SCIAMACHY and SAGE II measurements. Shaded areas show standard error of the mean.

$Ext_{750}(\alpha_{525/1020})$ are fairly constant with height and vary from 15 to 25% for $Ext_{525}$, from 10 to 25% for $Ext_{750}(PSD)$ and from 0 to 15% for $Ext_{750}(\alpha_{525/1020})$. The discrepancy between two different ways of computing $Ext_{750}$ derivation is quite remarkable, as with the consistent methods a better agreement is obtained. Though it should be highlighted once again, that for SAGE II 750 nm is not a measurement wavelength, and 525 nm is not considered in the SCIAMACHY retrieval. To

highlight the uncertainties coming from the different approaches to derive the $Ext_{750}$, we depict in Fig. 6 the mean relative differences between $Ext_{750}(\alpha_{525/1020})$ and $Ext_{750}(PSD)$ for the whole SCIAMACHY data set with a solid line, and $\pm 1$ the standard deviation shown as the shaded area. From Fig. 6 it is clear that the calculation of the extinction coefficient with Ångström exponent results in about 8% negative bias. This result is consistent with the result presented in the supplements to Rieger et al. (2015). Thus, it is important to consider this uncertainty when comparing the aerosol extinction coefficients from

different instruments measuring at different wavelengths.

## 5.2   Ångström exponents comparison with SAGE II

To extend the comparison to the data from SAGE II, $\alpha_{525/1020}$ were calculated from its aerosol extinctions, and compared to the ones derived from SCIAMACHY PSD product. The mean differences between SCIAMACHY and SAGE II Ångström exponents are presented in Fig. 7 with the solid blue line, with relative differences in panel (a) and absolute differences in panel (b). Following the previous comparison, the standard error of the mean is shown with the shaded area.

The relative difference between the instruments is altitude dependent, showing slightly different shape than $Ext_{1020}$. SCIA-MACHY $\alpha_{525/1020}$ is about 40% higher than that from SAGE II at 18 km, about 20% higher between 21 and 28 km, and the





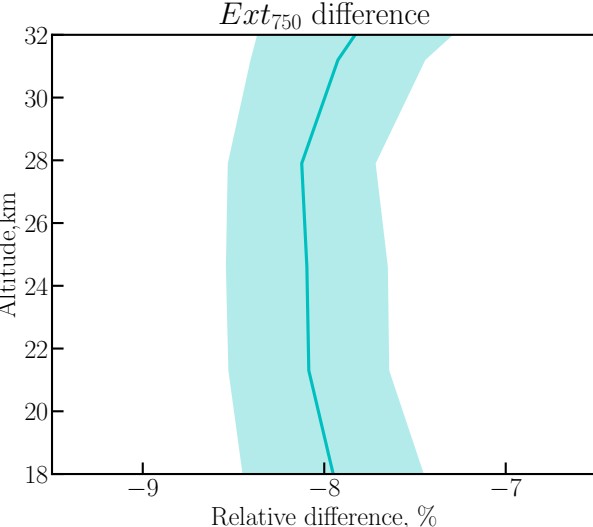

**Figure 6.** Mean relative difference between $Ext_{750}(PSD)$ obtained from PSD product with Eq. (3) and $Ext_{750}(\alpha_{525/1020})$ calculated using the Ångström exponent ($100 \times (Ext_{750}(\alpha_{525/1020}) - Ext_{750}(PSD))/Ext_{750}(PSD)$) from all available SCIAMACHY measurements. Shaded area shows $\pm 1$ standard deviation.

difference at 31 km is around 10%. In the absolute values the difference varies from 0.8 at the lowermost altitude to 0.2 at the uppermost one. The bias in this comparison is expected. As it was discussed in the previous section, the differences between SCIAMACHY and SAGE II for $Ext_{1020}$ and $Ext_{525}$ are both about 20%, but have the opposite sign, which results in the amplification of the error when calculating $\alpha_{525/1020}$. As for the $r_{eff}$ and $Ext_{1020}$, the reason why the difference between

SCIAMACHY and SAGE II is altitude dependent is still under investigation. To address this question the amount of collocations needs to be significantly increased to provide better sampling. This can be done by extending the SCIAMACHY PSD retrieval algorithm to other latitude bands and applying it to the profiles with cloud contamination.

### 5.3  Ångström exponents comparison with OSIRIS

Although comparison of SCIAMACHY with SAGE II provides some information on the agreement of the products, this

comparison is not sufficient to draw any robust conclusions. Additional information can be gained from the comparison with OSIRIS, which was operating at the same time as SCIAMACHY and also provided particle size information. Generally, comparison between SCIAMACHY and OSIRIS is more robust than with SAGE II, as both instruments employ the same measurement technique, use information at the same wavelengths (750 nm and 1530 nm), and provide a similar amount of measurements per orbit. Applying the same collocation criteria as for SAGE II, 4603 coincident profiles were found, which

is about half of the available SCIAMACHY profiles. The obtained amount of collocations is sufficient to ensure a reliable comparison. It is important to highlight once again that all the comparisons were done for the tropical region (20°N-20°S).





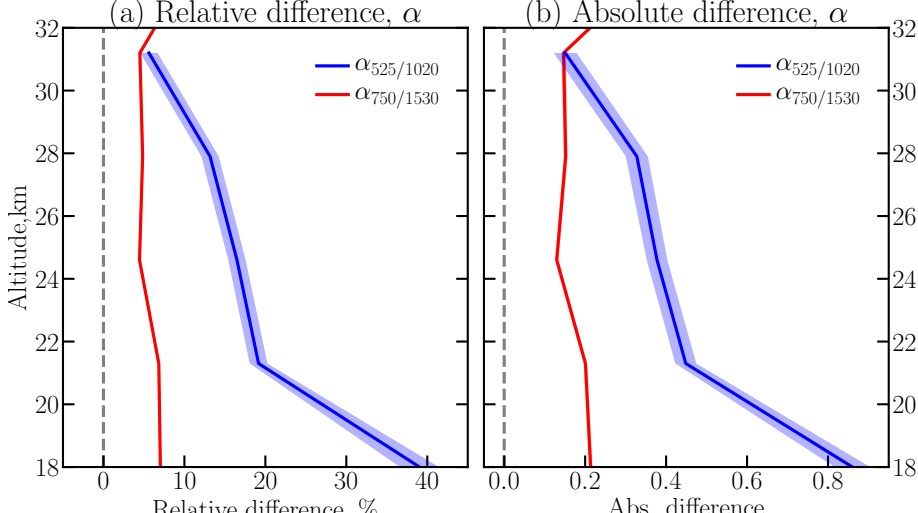

**Figure 7.** Mean relative (panel (a)) and absolute (panel (b)) differences ($200\times$(SCIAMACHY-instrument)/(SCIAMACHY+instrument)) between Ångström exponents from collocated SCIAMACHY and SAGE II ($\alpha_{525/1020}$: blue lines) and SCIAMACHY and OSIRIS ($\alpha_{750/1530}$: red lines) measurements. Shaded areas show standard error of the mean.

Differences between SCIAMACHY and OSIRIS $\alpha_{750/1530}$ are presented in Fig. 7 by the red line. OSIRIS $\alpha_{750/1530}$ was interpolated onto the SCIAMACHY measurement grid. The difference in vertical resolution was not accounted for as SCIAMACHY and OSIRIS have similar specifications. As for SAGE II, in panel (a) of Fig. 7 the mean relative differences are plotted with a solid line, while in panel (b) the mean absolute differences are depicted. The standard error of the mean is shown

by the shaded area; it is, however, within the thickness of the solid line. As follows from Fig. 7, the relative difference between SCIAMACHY and OSIRIS is about 7% for the lower altitudes and about 4% for the altitudes above 25 km. In absolute values the difference is about 0.2 below 25 km, and less than 0.15 at the higher altitudes. Taking into consideration the $\alpha_{750/1530}$ errors from SCIAMACHY (5%), estimated in Sect. 4.2, and the errors reported by Rieger et al. (2014) for OSIRIS Ångström exponents (10%), it can be concluded, that the Ångström exponents obtained from both instruments agree well with difference

being smaller than the reported errors.

To evaluate the temporal behaviour of the differences between SCIAMACHY and OSIRIS results the monthly zonal means of $\alpha_{750/1530}$ and its deseasonalized values (anomalies) at 21.3 km are plotted respectively in the panels (a) and (b) of Fig. 8. The deseasonalization of $\alpha_{750/1530}$ for the both instruments is justified, firstly, by the seasonality of stratospheric aerosols, which was reported in multiple studies including Hitchman et al. (1994); Bingen et al. (2004), and secondly, by the dependency of the

limb radiances on the seasonal changes of $SSA$. The deseasonalized values for each instrument were calculated individually by subtracting an averaged $\alpha_{750/1530}$ over all corresponding months in the whole observation period from each monthly mean value (e.g. the averaged $\alpha_{750/1530}$ for all the Julys from 2002 till 2012 was subtracted from the monthly mean value of each July



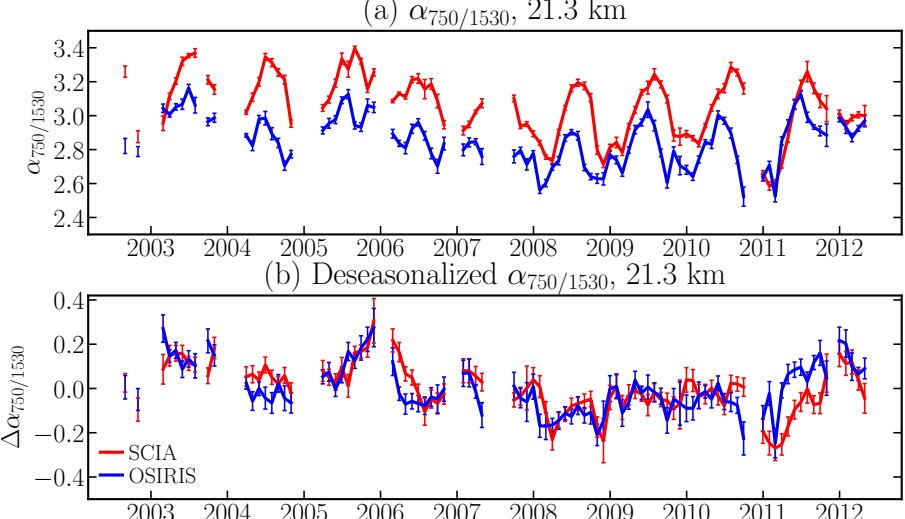

**Figure 8.** Zonal monthly mean (panel (a)) and deseasonalized (panel (b)) Ångström exponents ($\alpha_{750/1530}$) from collocated SCIAMACHY and OSIRIS measurements. Vertical bars show standard error of the mean.

in 2002-2012). Months with less than 10 collocations were excluded from the comparison. SCIAMACHY data is presented in red and OSIRIS in blue, the vertical bars show the standard error of the monthly mean value.

As seen in panel (a) of Fig. 8, the Ångström exponents retrieved from both instruments show very similar behavior, although the absolute values of $\alpha_{750/1530}$ from SCIAMACHY are systematically higher than those from OSIRIS. A high degree of 5 consistency is found between the results from both instruments in the comparison of the deseasonalized time series (see panel (b) of Fig 8). Generally, the blue and red lines overlap or lay within the error bars and follow the same pattern, except at the beginning of 2006, when SCIAMACHY values are slightly higher, and 2011, when OSIRIS is slightly higher. However, even in those periods the differences are rather small (about 0.05-0.08). As the differences between Ångström exponents from both instruments are fairly constant with the time, and do not show signatures of any remarkable events (e.g. volcanic eruptions), it 10 can be concluded, that they originate most probably from the technical specifications of the instruments and differences in the retrieval algorithms.

Summarizing Sect. 5, the following conclusions are made: aerosol extinction coefficients, obtained from SCIAMACHY PSD product agree with SAGE II within $\pm 25\%$, i.e. within the reported accuracy of the obtained $Ext$ from SCIAMACHY. The best agreement was acquired for $Ext_{750}$, calculated for both instruments from $Ext_{525}$ and $Ext_{1020}$ with $\alpha_{525/1020}$. 15 Additionally, it was shown that the recalculation of $Ext$ using the Ångström exponent can results in about 8% uncertainty. Ångström exponents from SCIAMACHY were compared to the ones from SAGE II and OSIRIS. The differences with respect to SAGE II results are about 40% at 18 km, decreasing to 20% at 21 km and to 10% at 30 km. With respect to OSIRIS, the agreement is much closer, with the relative differences between 4 and 7%. The differences between the instruments are smaller than the expected uncertainties, showing remarkable agreement between the instruments. The temporal behavior of $\alpha_{750/1530}$





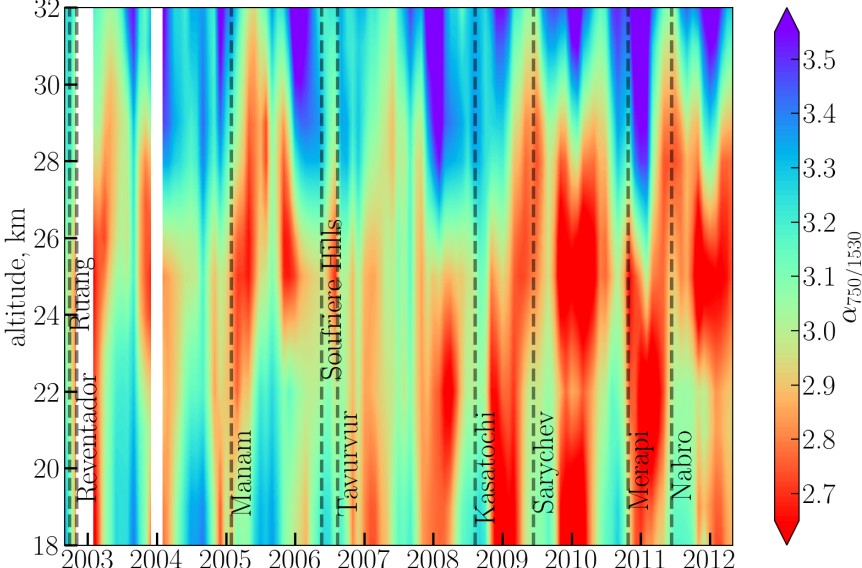

**Figure 9.** Monthly zonal mean values of the Ångström exponents ($\alpha_{750/1530}$) derived from SCIAMACHY limb data in the tropics (20°N - 20°S).

anomalies is independent of any remarkable events, with the minor differences coming from the small technical differences of the instruments and retrievals. The much better agreement of SCIAMACHY aerosol values with OSIRIS than with SAGE II is explained by a better consistency of the data sets: the same measurement technique and the same spectral information were used to retrieve the parameters.

## 6 Discussion

In order to investigate the temporal behavior of the Ångström exponent and to understand its dependency on the PSD parameters, the SCIAMACHY data set was analyzed. Unfortunately, due to rejection of cloud contaminated profiles temporal sampling of the obtained product is too sparse to analyze volcanic plumes. For this reason the monthly zonal mean (20°S-20°N) $\alpha_{750/1530}$ as shown in Fig. 9 were considered. With exception of the upper altitudes (26-32 km), where the quasi biannual oscillation (QBO) pattern is obvious, the seasonal variation of $\alpha_{750/1530}$ represents the dominating pattern seen in Fig. 9. As it was mentioned in Sect. 5.3, the seasonality of stratospheric aerosols was discussed in several previous studies. To make the analysis of the Ångström exponent behaviour after the volcanic eruptions more clear, $\alpha_{750/1530}$ was deseasonalized using the same approach as discussed in Sect. 5.3. The deseasonalized $\alpha_{750/1530}$ time series are presented in Fig. 10. It should be noted, that in Fig. 10 the increased Ångström exponent values are shown in blue, and decreased in red, as the increased $\alpha_{750/1530}$ is often interpreted as a decrease of the aerosol particle size, and vice versa.





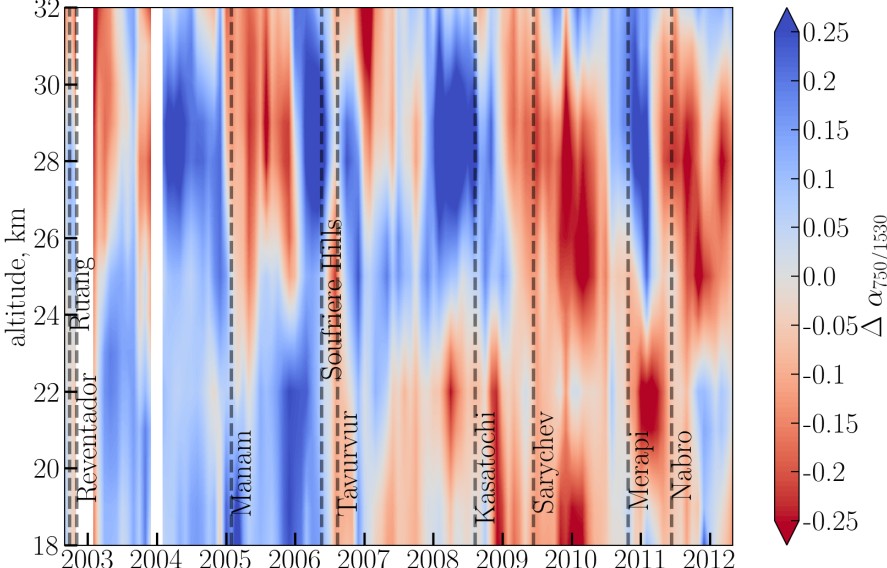

**Figure 10.** Deseasonalized time series (anomalies) of the Ångström exponents ($\alpha_{750/1530}$) derived from SCIAMACHY limb data in the tropics (20°N - 20°S).

Looking at Fig. 10, it becomes even more evident that the QBO pattern is well pronounced at altitudes above 26 km. This agrees with results reported earlier for SCIAMACHY aerosol products, e.g., Brinkhoff et al. (2015) showed similar patterns at around 30 km altitude for $Ext_{750}$. Later, the QBO signatures in $R_{mod}$ and $w$ were revealed by Malinina et al. (2018). The deseasonalized $\alpha_{750/1530}$ time series also demonstrate the influence of multiple volcanic eruptions. The slight decrease

of $\alpha_{750/1530}$ was noticed after the Tavurvur eruption, and more significant after the extra-tropical Kasatochi and Sarychev eruptions. Almost no change in $\alpha_{750/1530}$ was observed after Ruang, Reventador and Manam eruptions. After the Nabro eruption $\alpha_{750/1530}$ increases at the 18-23 km altitude. As for $R_{mod}$ and $w$ from the same data set (Malinina et al., 2018), the changes in $\alpha_{750/1530}$ reach higher altitudes with a certain time lag (tape-recorder effect). Interestingly, the general behaviour of $\alpha_{750/1530}$ looks very similar to that of $w$ (Malinina et al., 2018, Fig. 12). To evaluate it in more detail the dependency of

$\alpha_{750/1530}$ on $R_{mod}$ and $w$ was analyzed.

It is well known, that $\alpha_{750/1530}$ is dependent on both, $r_{med}$ and $\sigma$, but as $R_{mod}$ and $w$ are derived from these parameters, their impact on $\alpha_{750/1530}$ is not obvious. To investigate these relationships, the results from individual measurements in the tropical region over the whole observation period of SCIAMACHY are presented in Fig. 11. In panel (a), $\alpha_{750/1530}$ at all altitudes is presented as a function of $R_{mod}$ and $w$, while in panel (b) the dependency of $\alpha_{750/1530}$ on $r_{med}$ and $\sigma$ is shown.

The colors in Fig 11 depict the magnitude of $\alpha_{750/1530}$. From Fig. 11, it is clear that any particular value of $\alpha_{750/1530}$ can result from an infinite number of combinations of $R_{mod}$ and $w$ (or $r_{med}$ and $\sigma$). However we note that retrieving a pair of $R_{mod}/r_{med}$ and $w/\sigma$ is not the only way to obtain the Ångström exponent. As was discussed in (Malinina et al., 2018), in the spectral interval from 750 to 1530 nm the radiance can be fitted by two out of three PSD parameters, for SCIAMACHY the




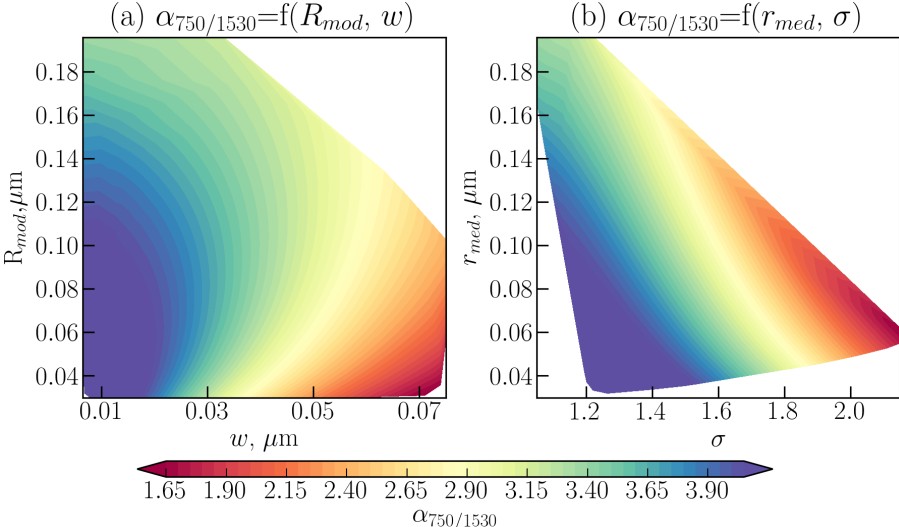

**Figure 11.** Dependence of the Ångström exponent ($\alpha_{750/1530}$) on mode radius, $R_{mod}$, and absolute distribution width, $w$ (panel (a)). In panel (b) $\alpha_{750/1530}$ as a function of median radius, $r_{med}$, and $\sigma$ is presented. Plot is based on the SCIAMACHY limb data in the tropics (20°N - 20°S).

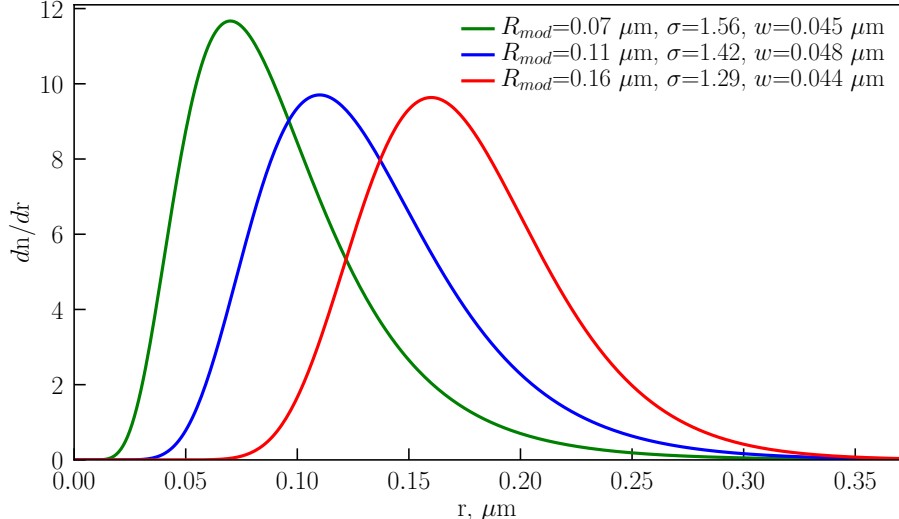

**Figure 12.** Aerosol particle size distributions with $\alpha_{750/1530}$=3.17. For convenience, $N$=1 cm$^{-3}$.

pair $R_{mod}$ and $\sigma$ was chosen as the limb radiances sensitivity to these parameters is higher than to $N$. However, it is possible to obtain a correct Ångström exponent also by retrieving, e.g., $r_{med}$ and $N$ (Rieger et al., 2014), although the accuracy of the PSD parameters may be not as high as for $R_{mod}$ and $\sigma$.




Comparing panels (a) and (b) it can be seen, that $\alpha_{750/1530} = f(R_{mod}, w)$ is a non-monotonic function, i.e. up to a turn-around point the same value of $\alpha_{750/1530}$ is obtained by increasing both, $R_{mod}$ and $w$, and then the same $\alpha_{750/1530}$ is a result of increasing $R_{mod}$ and decreasing $w$. The function $\alpha_{750/1530} = f(r_{med}, \sigma)$ is monotonic with respect to both, $r_{med}$ and $\sigma$, and there is the general rule: the larger is $r_{med}$ or $\sigma$, the smaller is $\alpha_{750/1530}$. The same value of $\alpha_{750/1530}$ can be reached by increasing $r_{med}$ and decreasing $\sigma$ or vice versa. It is important to highlight, that completely different distributions might result in the same value of $\alpha_{750/1530}$. To illustrate this fact, we chose three pairs of PSD parameters with $\alpha_{750/1530}$=3.17, and plotted the distributions $dn/dr$ in Fig. 12. The values of $R_{mod}$, $\sigma$ and $w$ used for the figure are listed in the legend. This figure disproves a widely spread belief, that smaller Ångström exponents are associated with the prevalence of larger particles and vice versa. As it can be seen, distributions with $R_{mod}$=0.07 $\mu$m and $w$=0.045 $\mu$m (green line) and $R_{mod}$=0.16 $\mu$m and $w$=0.044 $\mu$m (red line) have completely different amounts of large particles, but result in the same Ångström exponent. As it was already noted by Ångström (1929), $\alpha$ has only an "approximate coincidence with the average diameter directly measured". Thus, it can be concluded, that, firstly, there is no possibility to obtain a unique pair of PSD parameters from the known value of $\alpha_{750/1530}$, and secondly, to provide a relevant information on the change in the particle size, $\alpha_{750/1530}$ should be accompanied by one of the PSD parameter. Here, we note, that our conclusions are valid for the Ångström exponent at one wavelength pair. If several Ångström exponents for different independent wavelength pairs are provided, more information on PSD can be derived. However, for all known space-borne instruments providing aerosol information in the stratosphere only one value of Ångström exponent is reported in peer-reviewed publications, which makes our conclusions applicable to all of them.

Summarizing Sect. 6, it can be concluded, that $\alpha_{750/1530}$ can increase, decrease or remain unchanged after the volcanic eruptions. As for $R_{mod}$ and $w$ from the same data set, the tape-recorder effect after the volcanic eruptions as well as QBO signatures at upper altitudes (26-32 km) are observed. The pattern of $\alpha_{750/1530}$ changes is similar to that of the changes in $w$, although changes in both, $R_{mod}$ and $w$, contribute to the changes in $\alpha_{750/1530}$. It was also shown, that infinite amount of $R_{mod}$ and $w$ (or $r_{med}$ and $\sigma$) pairs result in the same $\alpha_{750/1530}$, and the statement that the large/small Ångström exponent means the prevalence of small/large particles is strictly valid only for increasing/decreasing $r_{med}$ with $\sigma$ remaining unchanged.

## 7 Conclusions

In this study, the stratospheric aerosol extinction coefficient and Ångström exponent have been discussed. From the investigation of the sensitivity of limb-scatter and solar occultation instruments to the aerosol particles of different size, it was shown that limb-scatter instruments are sensitive to the aerosol particles of smaller size, and thus, provide more accurate PSD information than solar occultation instruments, in particular, during periods with low aerosol loading. In contrast, occultation instruments provide aerosol extinction coefficients which are associated with smaller uncertainties than the ones from the limb instruments.

Here, we focus on the aerosol PSD product, which provides $R_{mod}$ and $\sigma$ (and recalculated $w$), obtained from SCIAMACHY limb measurements. In order to compare it with other space-borne instruments, aerosol extinction coefficients and Ångström exponents were recalculated using the PSD parameters. Error estimation based on the synthetic retrieval approach showed that the aerosol extinction coefficients are obtained with about 25% accuracy for the scenarios with high aerosol loading and with





less than 20% uncertainty for the background period. It was also shown that by using the retrieved $R_{mod}$ and $\sigma$ from the same data set it is impossible to estimate $N$, or put another constraint on the parameters by implementing coupled or a consequent $Ext$ retrieval. Ångström exponents calculated from the PSD parameters show less than 10% error for $\alpha_{525/1020}$ and less than 5% error for $\alpha_{750/1530}$. In the absolute values these errors are less than 0.2 and 0.15 respectively.

The recalculated aerosol extinction coefficients from the SCIAMACHY observations were compared to those from SAGE II. This comparison showed that differences are within $\pm 25\%$, which is within theoretically determined errors for SCIAMACHY. Ångström exponent ($\alpha_{525/1020}$) differences vary from 40% at 10 km to 10% at 30 km, with SAGE II values being systematically smaller. Furthermore, SCIAMACHY Ångström exponents ($\alpha_{750/1530}$) were compared to those from OSIRIS (another limb-scatter instrument). The relative difference between the instruments is decreasing from 7% at the lowermost altitudes to 4% at

the uppermost altitudes. The absolute values of $\alpha_{750/1530}$ differ by less than 0.2, and both relative and absolute differences are within the theoretically determined errors of $\alpha_{750/1530}$ for those instruments. The time series analysis of the collocated data sets showed that the differences do not change significantly with time and are not correlated with any remarkable events, such as volcanic eruptions.

The Ångström exponent in the tropics was analyzed using the values, recalculated from SCIAMACHY PSD data set. It

was shown that the monthly $\alpha_{750/1530}$ anomalies show distinct QBO signatures in the upper stratosphere, and the Ångström exponent can either increase, decrease or remain unchanged after a volcanic eruption. The analysis showed that changes in $\alpha_{750/1530}$ are driven by changes in both $R_{mod}$ and $w$ (or $r_{med}$ and $\sigma$), and an infinite number of pairs of these parameters provides the same value of $\alpha_{750/1530}$. It was concluded, that it is impossible to derive any reliable information on the changes in the aerosol size based solely on the Ångström exponent for one wavelength pair. This can only be done if at least one of the

PSD parameters is provided in addition.

*Data availability.*  SCIAMACHY aerosol particle size distribution data is available after registration at http://www.iup.uni-bremen.de/scia-arc/. OSIRIS data set can be downloaded at http://odin-osiris.usask.ca/?q=node/280.

*Competing interests.*  The authors declare that they have no conflict of interest.

*Acknowledgements.*  This work was funded in parts by European Space Agency (ESA) through SQWG and SPIN projects, German Aerospace

Center (DLR) through SADOS project, German Federal Ministry of Education and Research (BMBF) through the ROMIC-ROSA project, the Natural Sciences and Engineering Research Council (Canada), the Canadian Space Agency, University and State of Bremen. The Post-graduate International Programme in Physics and Electrical Engineering (PIP) and University of Bremen provided a grant for Elizaveta Malinina to visit the University of Saskatchewan. We thank ECMWF for providing pressure and temperature information. We also thank NASA for SAGE II data which was downloaded from the NASA Langley Research Center EOSDIS Distributed Active Archive Center. The

manuscript is a part of preparatory studies for DFG Research group project VolImpact.



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
