# Peer review of "Stratospheric aerosol characteristics from space-borne observations: extinction coefficient and Ångström exponent"

_Atmospheric Measurement Techniques, 2018_

## Referee Comment (RC1) · Anonymous Referee #2 · 27 Nov 2018

General comments on Malinina et al. (2018):

This paper provides a useful framework for discussion of the information content of limb scattering vs. occultation measurements. The abstract states that "limb instruments have better potential for the PSD retrieval." This statement appears to be true, primarily because (as shown in the text) the scattered radiances are more sensitive to smaller particles than the transmitted radiation measured by occultation instruments. However, this conclusion is somewhat weakened by the last sentence of the abstract, which (correctly) reports that the retrieved quantity (Angstrom coefficient for a single pair of wavelengths) could correspond to an infinite number of combinations of the PSD

parameters. Two basic problems arise that deserve greater consideration:

1. The analysis presented throughout this paper assumes that the PSD has a single-mode log-normal shape. So the analysis shows that a set of median radius + mode width pairs could produce the same Angstrom coefficient, but confining all analysis to the single-mode log-normal possibility understates the ambiguity that actually exists: Many other types of PSD functions (bi-modal, gamma distribution, innumerable other functional shapes) exist that could also produce a given Angstrom coefficient.

2. Some work is cited that uses extinction measurements to infer PSD information, simply by assuming a type of PSD and selecting one of the many sets of PSD parameters that are consistent with the observations + the spectral variation derived from Mie theory (Yue, 1999, for example). In many respects, the same method is used here for limb scattering, and it is not clearly quantified how the approach leads to better results when limb scattering measurements are used (granting that this is likely to be true because, once again, the scattering measurements are more sensitive to smaller particles).

Specific comments:

Abstract:

The term "remarkable events" seems to be used as a synonym for "volcanic eruptions that perturb the stratosphere" (here and in several subsequent places). This should be clarified, perhaps by replacing that phrase with something more specific (such as "volcanic perturbations?")

Sect. 1, 2nd paragraph:

The fact that the aerosols are assumed to be spherical should be explicitly stated here.

Sect. 1, 3rd paragraph:

The second sentence ("Known existing . . . (Damadeo et al., 2013)" is awkwardly

worded.

Sect. 2.1, 1st paragraph:

The solar/lunar occultation mode of the SCIAMACHY instrument is mentioned here – are those observations usable for this study? If the measurements have sufficient quality, they would certainly add to the limited number of coincident occultation + limb scattering measurements.

Sect. 3.1, 1st paragraph:

As discussed earlier, the failure to consider the consequences of any PSD other than single mode log-normal is a significant limitation of this study (just as it is a significant limitation of most limb scattering aerosol retrieval work to date). Even a single example testing this approach for another type of PSD (realistic, but not single mode log-normal) would add significant value to this study, by providing some indication of how much the results presented depend on that restriction.

Sect. 3.2, 5th paragraph:

It would be helpful to provide some guidance about the sigma value that corresponds to w = 0.01 microns for a few examples, since this is an unconventional way to describe the PSD.

Sect. 3.2, 7th paragraph:

The basis for the perturbation analysis presented is confusing: Apparently the temperature and pressure are treated as variables that can be perturbed independently (without regard for hydrostatic balance, for example)?

Figure 9:

This figure illustrates some limitations on the conclusions that can be drawn from this analysis due to the constraints that have been imposed on the PSD during this analysis: What does it mean to perform a retrieval based on a single PSD, then analyze the

variation of Angstrom coefficient with altitude (which should be zero, if a single PSD truly characterizes the stratospheric aerosol)? Or am I misinterpreting something in the methodology?

References:

Yue, G. K.: A new approach to retrieval of aerosol size distributions and integral properties from SAGE II aerosol extinction spectra, J. Geophys. Res., 104, 27 491–27 506, 1999.

---

## Referee Comment (RC2) · Anonymous Referee #1 · 10 Dec 2018

**Referee Report on "Stratospheric aerosol characteristics from space-borne observations: extinction coefficient and Angstrom exponent, by Malinina et al.**

**General comments:**

This paper reports the continuation of a previous work published by Malinina et al. (2018) presenting a new dataset of particle size information retrieved from SCIAMACHY. It proposes an extension of the dataset including the Angström coefficient and the aerosol extinction coefficient in the tropical zone (20°S-20°N), as well as a sensitivity study and error analysis of their dataset. Afterward, the authors explore the link between the Angström coefficient and the parameters characterizing the particle size distribution (PSD). Based the sensitivity study, the authors claim that limb viewing instruments are more accurate for PSD retrieval than occultation instruments.

If the approach chosen by the authors provides an interesting and valuable insight into the problem of aerosol size retrieval from spaceborne instruments, I cannot agree at all with their conclusions on the sensitivity study. This conclusion is based on a too limited and biased analysis, which doesn't take (sufficiently) into account critical elements such as the impact of the bias induced by the assumptions made on the PSD in the forward model, and the influence of the covered spectral range and of the use of multiple wavelengths on the information content.

Further, the authors propose a long discussion about the usability of the Angström coefficient to derive size information and about the ill-posedness of the problem. Beyond long developments that are in some cases not really new or relevant, many arguments used in this discussion are truncated or even wrong, and this study should benefit a lot from a more accurate reading of the literature on this subject.

If the authors want to compare the capabilities of occultation and limb viewing experiments, they need to revise thoroughly their sensitivity study to take into account all aspects of the inversion problem, including what was published in the past on this topic. As presented here, most of the conclusions of the authors, including the conclusion that limb-viewing instruments are more accurate than occultation instruments cannot be drawn, and are thus basically wrong.

**Detailed comments:**

Abstract:

- L. 5-6, p.1, "These uncertainties can be mitigated...": I am not sure that the mitigation is very efficient, because the assumptions made on the PSD for the forward model obviously precede the PSD retrieval, thus influences the PSD retrieval.
- L. 6-8, p.1: I don't agree with this statement. See discussion later on the body of the manuscript.

- L. 12-14, p.1: This is not a correct and complete estimate of the error on the extinction and Angstrom coefficient. This statement has to be revised. See comments on Section 5.1.
- L. 16-17, p.1: Since OSIRIS is based on the same measuring technique (limb viewing geometry) and makes use of very similar assumptions on the PSD as SCIAMACHY in the forward model, the results of this comparison have to be considered very cautiously. This should be mentioned by the users. See also remark on Section 5.3.
- L. 19-21, p.1: Also, the Angström coefficient depends on the considered spectral range. This should also be mentioned.

**1. Introduction:**

- L. 4-5, p.2: I don't understand this sentence: the role of stratospheric aerosols on what ? There is an abundant literature about stratospheric aerosols, and stratospheric aerosols are the scope of a SPARC (Stratospheric Processes and their Role in Climate) activity called SSiRC (Stratospheric Sulfur and their Role in Climate, See Kremser et al., 2016). The authors mention many works addressing the role of aerosols in climate, in the specific case of Asian summer monsoon, in geoengineering, and many other aspects. The aerosol evolution and role in the radiative forcing of the atmosphere was also the subject of several publications (e.g. Bingen et al., Remote Sens. Env., 2017; Brühl et al., Atm. Phys. Chem., 2018). On the other hand, it is true that the current period is characterized by a particularly low amount of satellite experiments with profiling capabilities. If this is what the authors want to (rightly) emphasize, they should reword their sentence. Otherwise, they should remove this sentence.
- L. 18, p.3: GOMOS processed with the AerGOM algorithm provides the extinction coefficient in the range 300-750 nm (Vanhellemont, et al., 2016, op. cit.).
- L. 18-20, p.3: The conversion of the basckatter coefficient to extinction is not a straightforward process since it requires the knowledge of the lidar ratio, a time-, space- and aerosol composition-dependent parameter. Errors on this parameter can thus induced a large variability and a large uncertainty on the derived extinction. This fact should be mentioned in the present discussion. Several publications address this problem, e.g. Rogers et al., Atm. Meas. Tech, 7, 4317-4340, 2014.

**2. Instruments and data:**

- L. 15-16, p.4: how was the fixed particle number density determined from ECSTRA ? ECSTRA is a climatology of stratospheric extinction based on a parameter describing the overall volcanic state of the atmosphere, but does not provide size parameters. Please describe your methodology.
- L. 20, p.5: It should be mentioned that the approach proposed by Thomason et al, 2008 concerns the non-volcanic case.

**3. Sensitivity of measurements to aerosol parameters**

- L. 26-27, p.5: "unimodal" and "lognormal" are two independent concepts. The authors should add something like: ", here with a lognormal function:"
- L. 28, p.5: It might be useful to mention units, especially for the particle number density.
- L. 6-7, p.6: I don't understand this statement. Several degrees of freedom are required to retrieve the extinction coefficient at several wavelengths. It may look like extinction retrieval requires less degree of freedom than PSD retrieval, but, as explained at the end of the section, this is due to the fact that a very significant information content is hidden in the model used, more particularly in terms of PSD assumed in the forward model, including a distribution function and the related mode parameters. Consequently, the authors should qualify this statement, make the link with the important reminder at the end of the section, and at least precise if they are only considering the case of limb viewing instruments for which additional information on the PSD is provided in the forward model.
- L. 8-9, p.6: The formulation of the extinction coefficient is not correct. It has to be written as the integration of the PSD weighted by the extinction cross-section (See, for instance, d'Almeida et al., Atmospheric Aerosols, Deepak Publishing, 1991). Eq. (3) amounts to considering that the extinction cross-section can be approximated by its value for a fixed particle radius $r_{med}$ and a wavelength $\lambda$, and put out of the integral, what in general is not correct. The extinction cross-section is a parameter describing the extinction of light with wavelength $\lambda$ by a single particle characterized by its radius and index of refraction. Hence, it doesn't depend on aerosol mode parameters. It is very important to clarify how $\beta_{aer}$ is computed because it determines how to interpret the error assessment for the extinction in §4.1.
- L. 6-8, p.7: This is confusing. Are the authors talking about the occultation case (Eq. (5))?
- L. 11-13, p.7: Since the solar scattering angle influences the phase function which depends critically on the PSD (the phase function being a weighted integration of the PSD), this issue has to be carefully investigated.
- Eq. (7) and (8): The parameter K is different in both equations and has actually different dimensions in both cases. Hence, a different notation should be used, for instance, $K_{obs}$ and $K_{scat}$.
- L. 23, p.7: There is a factor $\pi$ missing in the expression of $K$, related to the particle cross-section $\pi r^2$.
- L. 26, p.7: "Showed it": What did they showed?
- L. 25-27, p.7: I am not sure that these details are useful: it seems that Thomason's formulation is different although it is the same, admittedly based on another choice of variable (volume of aerosol per volume of air, instead of particle radius). The authors might consider just mentioning that Thomason and Poole use a similar formulation instead of emphasizing the differences, in order to avoid confusion.
- L. 1, p.8: There seems to be an error in $K$'s dimensions in the case of OSIRIS. If Eq. (7) is applied, $K$'s units should be expressed in W (with $I$ in W/m$^2$ and $n(r)$ in

number of particles per volume unit). Even if applying the expression derived by Rieger et al. (2014), a radiance factor appears in the expression. Consequently, $K$ should include the contribution of the energy flux, and not be dimensionless.

- L.14, p7-L. 11, p.8: Overall, this discussion is a bit confusing: It seems that the aim is to show that in all cases, the inversion problem can be formulated using a similar expression, either using Eq. (7) for the occultation case, or Eq. (8) for the limb case. But immediately afterward, it is explained why this model is considered as much too simple in the OSIRIS case, and how it is not suited and will not be used for sensitivity studies in the case of SCIAMACHY. What is then the utility of this discussion?
- Eq. (9), p.8: This separation between "aerosol" and "Rayleigh" signals supposes that there is a clear distinction between both, and the authors most probably infer this Rayleigh signal from ancillary data of air density, temperature and pressure. The reality is much more complex, since very thin aerosol particles are also Rayleigh particles and their contribution to scattering cannot be discriminated from the molecular Rayleigh compound. This is especially the cases for "thin aerosol" cases considered by the authors. The only way to separate both contributions is to rely on the meteorological data that might be inaccurate with respect to the local condition encountered by the spaceborne instruments.
- L. 5-11, p.8: This model only takes into account the Rayleigh and aerosol compounds. How do the other contribution (trace gases) interfere in this analysis ?
- L. 12-13, p.8: After the previous discussion, we know that Eq. (8) will not be used, but not which model/expression/equation was actually used to quantify the sensitivity. This should be clarified.
- L. 17, p.8-L. 14, p.9: This sensivity study is limited to the sensitivity of individual extinction channels, and this for two wavelengths in the infrared, including the 1530 nm-wavelengths, which is much higher than the particle size range the authors consider as relevant (50 -300 nm). The behaviour of the sensitivity curve S and its quantitative assessment is absolutely insufficient to assess the performances of (existing) limb viewing instruments versus occultation instruments, for several reasons:
  - If it is true that the size distribution influence twice the expression of the radiance (Eq. 6, through the scattering coefficient and the phase function), the error made by assigning inaccurate values of the aerosol mode parameters affects equally twice this parameter.

[Figure]

Efficiency curve for m = 1.435

o   The value of the wavelength influences significantly the scattering
    efficiency, as illustrated on the figure above. In particular, if the wavelength
    is very large compared to the particle radius, aerosol particles behave as
    Rayleigh particles, scattering becomes independent of the particle size
    (scattering $\sim \lambda^{-4}$) and the size parameter cannot be discriminated. The
    figure shows the dependence of the scattering efficiency as a function of the
    parameter $x = 2\pi m / \lambda$ (see e.g. van de Hulst, Light Scattering by Small
    Particles, Dover Publications, 1957), where the refraction index m is
    representative for a 75% $H_2SO_4$-25% $H_2O$ aerosol composition and the
    spectral range 350-1530 nm. It also shows, for particle size range 50-300
    nm used on Figure 1 of the paper, the range of x parameter covered for the
    two wavelength considered by the authors ($\lambda$ = 750 and 1530nm, in red and
    magenta) and two other wavelengths representative for the spectral range
    covered by the most occultation instruments observing in the UV-visible-
    near IR range($\lambda$ = 350 and 500 nm, in cyan and green). It is obvious that the
    dynamic range corresponding to 1530 nm is particularly reduced (and
    similar to the Rayleigh regime). The 750 nm channel provides more
    variability in the scattering efficiency curve than the one at 1530 nm, but
    with a large overlap with the dynamic range of this first channel. On the
    other hand, wavelengths of 500 nm and 350 nm provide a much larger
    dynamic range covering almost all possible values of the scattering
    efficiency between 0 and more than 4 (van de Hulst, op. cit., 1957).

o The authors don't take at all the critical aspect that particle size retrieval can be retrieved by the combination of extinction values at several values. In this respect, it is clear from the figure above that the dynamic range covered by the wavelengths spread over the UV-visible-near IR range like for instruments such as SAGE II and III, POAM III, or GOMOS provides a much larger information content than the combination 750-1530 nm. This aspect is of crucial importance in the case of real aerosol particle population where the diversity of particle sizes blurs the scattering response of individual particles, especially when several aerosol modes are present simultaneously (i.e. with a large mode width of the equivalent lognormal size distribution). The larger the spectral range covered in the Mie regime, the higher the information content.

As a conclusion of this discussion, even if the calculation of the modelled sensitivity is correct, it is not sufficient to assess the performances of a technique and it doesn't provide definitive arguments to conclude about the comparison of the overall capabilities of (existing) limb viewing versus occultation instruments.

- L. 8-9, p.9: 0.1 µm is not a magic limit for the sensitivity of occultation instruments, but it corresponds to some upper limit of the Rayleigh scattering regime. It depends thus on the spectral range covered by the instrument.
- Figure 2, p.9: Both figures compare the limb and occultation geometries in a spectral range limited to wavelengths values where the occultation geometry is particularly insensitive, and that doesn't correspond to the spectral range covered by most occultation instruments (See above). It is actually visible that the sensitivity in the occultation case is increasing toward the smallest wavelengths. Consequently, this comparison is biased and it doesn't reflect the true sensitivity of real sensors used for aerosol remote sounding from space.
- L. 19-25, p.10: Taking into account the various elements cited above, I cannot agree at all with these conclusions.

**4. Error assessment**

- L. 15-17, p.11: The chosen scenarios cover quite nicely aerosol particle populations with radii up to about 250-300 nm. One should still bear in mind that volcanic aerosols, in view of existing estimates from satellite and balloon-borne datasets, may reach significantly higher values during particular (volcanic) periods the SCIAMACHY lifetime.
- L. 15-22, p.11: If I understand well, the errors (calculated as the "median relative error") reflects the variability obtained from an ensemble of simulations where Gaussian noise was added to the synthetic radiance. On the other hand, the synthetic scenarios are constructed using a fixed size distribution (i.e. a fixed choice of $r_{med}$ and $\sigma$) and only N is supposed to decrease exponentially as described in L. 9, p.11. The realistic character of this profile is thus very relative. Consequently, the quantity investigated here is very different from the extinction uncertainty, that include experimental, instrumental,

modelling and other retrieval errors into account.  The use of the term "error" is thus particularly misleading, and should be replaced by something more adapted.

- L. 1-4, p.12: See remark on L. 8-9, p6.
- Eq. (10), p.13: See remark on L. 8-9, p6.
- L. 22, p.13-end of p.14: Same remark as for  L. 15-22, p.11.

**5. Comparison of the measurements results**

- L. 10-11, p.15: The difference in the measurement technique is not an issue at all ! It is the essence of validation efforts to use different datasets, including datasets based on different measurements techniques, to assess the strengths and weaknesses of the measurements to be investigated. This statement is thus inappropriate and should be removed.  Arguments developed here may be pertinent to discuss the origin of possible weaknesses and strengths, but not as some kind of a priori disclaimer to relativise the adequateness or validity of a comparison exercise.
- L. 12-13, p. 15: As explained above, the sensitivity analysis proposed here is biased and insufficient, mainly because it doesn't take into account the whole spectral range taken covered by the occultation instrument, here SAGE II. Hence, even if it has been shown, indeed, that SAGE II is less sensitive to thin particles than to particles in the range ~0.25-0.40 μm (See for instance Bingen et al., Ann. Geophys., 2003 for a comparison with balloonborne measurements) so that $r_{eff}$ is indeed expected to be biased high in the case of SAGE II, a more rigourous analysis taking into account all aspects of measurements and retrieval (including the impact of the different assumptions and approximations made in the limb viewing case) is needed to draw definitive conclusions about comparisons between SCIAMACHY and SAGE II.
- L. 28, p.15: "the standard error on the mean relative difference" might be more clear.
- L.28-29, p.15: Both parameters provide different information, and the one presented should be the most appropriate to assess the quality of the agreement between datasets ! If the standard deviation is so large that it makes the figure very busy, it means that the quality of the profile-to-profile comparison is very poor, and this should also be shown in some way !
- L. 33-34, p.15: I don't see why a similar behaviour is expected for the mean relative difference of the extinction at 1020 nm and $r_{eff}$: the extinction at 525 nm, which shows a quite different behaviour on Figure 5, plays an equally important role in the computation of SAGE II's effective radius (Thomason et al., 2008, op. cit.).
- L. 34, p.15: What is the usefulness of the reference to Sect. 2.3 ? There is nothing more about $r_{eff}$ is Sect. 2.3 than what is said here. A reference to some paper where the derivation of $r_{eff}$ is presented would be more useful.
- L. 4-10, p.16: This figure only shows that there is a bias of about 8% between the two ways used to derive the extinction coefficient from the PSD. This is very different from providing any assessment of the error on the extinction

coefficient. Further, the uncertainty on the PSD has, to my knowledge, not been correctly characterized: available "errors" (Malinina et al., op. cit., 2018) are derived in a similar way as the extinction "error" in the present paper and similarly express a variability with respect of an ensemble of more or less realistic synthetic cases (See remark on L. 15-22, p.11). Hence, I am not very sure Figure 6 adds more information on the extinction uncertainty, other than emphasizing some incoherence in the processing chain producing Ext, the PSD and the Angström exponent, most probably related to the successive assumptions and approximations made.

- L. 16, p. 16: What do the authors mean ? The green curve corresponding to the $Ext_{1020}$ in Figure 5 and the blue curve corresponding to $\alpha_{525/1020}$ have completely different shapes !
- L. 2, p. 17: What do the authors mean by "the bias in this comparison" ? I guess they just mean something like "this behaviour" ?
- L. 5-7, p. 17: I am not sure that adding complexity to the problem by extending the latitudinal range will help answering this question. A better way is probably to carefully examine the impact of every assumption and approximation made in the SCIAMACHY retrieval, and its possible altitude dependent character. Using real and totally independent measurements like balloonborne OPC measurement profiles to reconstruct a forward model and every step of the retrieval might help understanding the remaining issues.
- L. 15, p.19: I don't agree with that. As explained above, 8% is the difference that was used between the two methods used here, but it does not represent the uncertainty on the recalculated $Ext_{750}$. What can be concluded from this calculation is that the additional uncertainty on $Ext_{750}$ due to the recalculation is at least 8%.
- L. 1-4, p.20: It is very likely that, beyond the same measurement technique and the same use of spectral information, the similarities between the retrieval schemes, including the assumptions and approximations made, greatly contribute to the similar performances on both datasets. Hence, both SCIAMACHY and OSIRIS datasets might present similar biases, which are impossible to discern from the comparison SCIAMACHY-OSIRIS but potentially (very) significant. In this way, the comparison might be (very) "unfair" with respect to SAGE II and lead to the possible wrong conclusion that SCIAMACHY and OSIRIS are likely to be more accurate than SAGE II. It is very important to add such a discussion point here to exclude wrong conclusions on the respective degree of reliability of SCIAMACHY, OSIRIS, and SAGE II.

**6. Discussion**

- L. 16, p.21-L. 3, p.22: I don't understand the rationale used here. We are dealing here with an underconstrained problem (as stated by the authors in L. 15-16, p.21) with 3 unknown: some radius and some spread characterizing the shape of the PSD, and a measure of the number of particles characterizing its amplitude. The way used to correctly constrain the problem was to fix the particle number for Malinina et al., and to fix the spread for Rieger et al. These

are just two possible choices (amongst other possible ones), and combining the results of both approaches will not bring any additional information at all ! Further, using {$R_{mod}$, w} or {$r_{med}$, $\sigma$} are just two equivalent ways to model the same thing, as nicely illustrated by the two panels in Figure 11.

- L. 1-17, p.23: If I understand well, the authors found that there are an infinite number of solution for a PSD giving a the same spectral dependence of the extinction as one value of the Angström coefficient. Since 3 parameters are used in the PSD and only one for the Angström coefficient to describe the same kind of information (actually the spectral dependence of the measured signal), this result is quite trivial and I don't see the added value of such a long discussion. It would be much more interesting to explain how the authors intend to solve the problem in the case of SCIAMACHY. The way to deal with this problem was already addressed in the past (Echle et al., J. Geophys. Res., 103, 19993-19211, 1998; Fussen et al., Atmosph. Env., 35, 5067-5078, 2001), so that, if the authors are willing to present their own method applied to SCIAMACHY, they should at least refer to these works in the discussion.

- L. 15-18, p.23: This part of the discussion are absolutely truncated. The authors are here finding that more than one Angström coefficient, or equivalently, more than 2 extinction channels, should help deriving the PSD. There is absolutely nothing new in this, and telling that "all known space-borne instruments" provides only one Angström coefficient amounts to saying that all known space-borne instruments provides extinction at, at most, two wavelengths, what is obviously wrong. In particular, as mentioned above, extinction coefficients from occultation instruments are used over several wavelengths covering a large spectral range, what helps solving the ill-posedness of the problem (see e.g. Fussen et al., Atmosph. Env., 35, 5067-5078, 2001; Bingen et al., Ann. Geophys., 21, 797-804, 2002). Actually, this disappointed findings shows again that the comparison made here between occultation and limb viewing instruments is very incomplete, and the statement that limb instruments have a better potential or are more sensitive to the aerosol size is wrong or, at least, very premature.

- L. 18-23 p.23: After an eruption, the PSD evolves according to the microphysical processes taking place, and using a Mie model, the evolving PSD determines unequivocably the extinction behaviour, and hence the Angström coefficient through Eq. (10). The inability to derive unambiguously the PSD from the Angström coefficient obviously doesn't implies the other way around. The authors seem to claim that the behaviour of the Angström coefficient is unpredictable and depends on the detail of their formalism ("with $\sigma$ remaining unchanged"), but this is obviously wrong and the point is that the exploration of Figure 11 just doesn't provide sufficient information to foresee the behaviour of the Angström coefficient after an eruption.

**7. Conclusion**

- Based on all what was explained above, I disagree on most of the conclusions given here. The reasons for this have been given before and don't require any

repetition. I would just mention that the "most correct conclusion" in my opinion (L. 18-19, p.24: "it is impossible to derive any reliable information (…)") is still incorrect: the information provided by the Angström coefficient is incomplete, but is reliable if the extinction measurements used for its calculation are reliable.

**Technical corrections:**

- L. 11, p.2: "periods of heavy aerosol loading" ?
- L. 34, p.2: Incorrect use of the parentheses; reference "Bingen et al" could be moved after "Known existing datasets" to make the text more fluent: "Known existing PSD data sets (Bingen et al. (2004)) were obtained (…) to 2005 (Yue et al., 1989; Thomason et al (2008) etc.).
- L.30, p.3: "errors in the extinctions" or "errors in Ext".
- L. 24, p.5: "following a lognormal distribution".
- L. 7, p.6: "by lack of information"?
- L. 6, p.7: Incorrect sentence.
- L. 15, p.9: duplicated "the".
- L.3, p.10: "reasonable".
- Caption Figure 3: "The solid lines show the extinction calculated from PSD…" ?
- L. 7, p.12: "latitudinal".
- L.8, p.13: "a subject for further studies" ?
- L. 28, p.15: The authors might add the colour used for $Ext_{750}(\alpha_{525/1020})$ to be complete.

---

## Author Comment (AC1) · 31 May 2019

We thank the reviewers for the time they spent thoroughly reading the manuscript and commenting on the paper. We hope we have answered the reviewers' questions and improved the explanations where needed. To distinguish the referees' comments from the author's responses, the comments are shown in italicized font and the responses are highlighted in blue.

*General comments on Malinina et al. (2018):*
*This paper provides a useful framework for discussion of the information content of limb scattering vs. occultation measurements. The abstract states that "limb instru-*

[Figure]

*ments have better potential for the PSD retrieval." This statement appears to be true, primarily because (as shown in the text) the scattered radiances are more sensitive to smaller particles than the transmitted radiation measured by occultation instruments. However, this conclusion is somewhat weakened by the last sentence of the abstract, which (correctly) reports that the retrieved quantity (Angstrom coefficient for a single pair of wavelengths) could correspond to an infinite number of combinations of the PSD parameters.*

Based on this and next comments of the reviewer we realized, that there might be a misinterpretation of the general concept of our research. In our method, we used the retrieved PSD parameters: $R_{mod}$ and $\sigma$ (the retrieval method was presented in Malinina et al. (2018)); and then from the retrieved PSD parameters extinction coefficients and Ångström exponents were recalculated with Mie theory. We understand that this misinterpretation was caused by our formulations, so the text of the revised manuscript has been changed accordingly.

*Two basic problems arise that deserve greater consideration.*
*1. The analysis presented throughout this paper assumes that the PSD has a single-mode log-normal shape. So the analysis shows that a set of median radius + mode width pairs could produce the same Angstrom coefficient, but confining all analysis to the single-mode log-normal possibility understates the ambiguity that actually exists: Many other types of PSD functions (bi-modal, gamma distribution, innumerable other functional shapes) exist that could also produce a given Angstrom coefficient.*

We agree with the reviewer, that there is an ambiguity in the assumption of the PSD shape only by one type, and in the reality the shapes might change from one event to another. However, for our PSD parameters retrieval (see Malinina et al. (2018)), the uni-modal log-normal distribution is assumed. The study presented here is based on

this particular product and, thus, is related to this particular assumption, which for now haven't been proven to be either better, or worse than the other ones, suggested by the reviewer. For that reason, we find it unnecessary to conduct studies with the other assumptions, as completely new retrieval algorithms need to be developed for this purpose, and there is not enough information to retrieve, e.g., bi-modal distribution. We changed the text of the manuscript to highlight, that our conclusions are based on our particular product and particular assumption.

*2. Some work is cited that uses extinction measurements to infer PSD information, simply by assuming a type of PSD and selecting one of the many sets of PSD parameters that are consistent with the observations + the spectral variation derived from Mie theory (Yue, 1999, for example). In many respects, the same method is used here for limb scattering, and it is not clearly quantified how the approach leads to better results when limb scattering measurements are used (granting that this is likely to be true because, once again, the scattering measurements are more sensitive to smaller particles).*

As mentioned in the first reply to the reviewer, there is a misinterpretation of the study concept. We do not use the method, which is usually applied for occultation measurements. In the previous paper (Malinina et al., 2018), which was cited in the manuscript, we state, that SCIAMACHY PSD retrieval does not use the retrieval of the aerosol extinction coefficient as an intermediate state. This particular retrieval algorithm retrieves $R_{mod}$ and $\sigma$ from the limb radiances directly. For this reason, we consider this reviewer's comment as inapplicable. The misinterpretation was caused by poor wording of the original manuscript. To make the text of the revised manuscript more clear, we added a more detailed description of the algorithm, and highlighted that the study is based on the PSD product.
*Specific Comments:*
*Abstract:*
*The term "remarkable events" seems to be used as a synonym for "volcanic eruptions that perturb the stratosphere" (here and in several subsequent places). This should be clarified, perhaps by replacing that phrase with something more specific (such as "volcanic perturbations?")*

By the remarkable events here and further not only the volcanic eruptions were meant, but also biomass burning events (e.g. Black Saturday in 2009) or any maintenance works or degradation of the instruments. We clarified that in the revised manuscript.

*Sect. 1, 2nd paragraph:*
*The fact that the aerosols are assumed to be spherical should be explicitly stated here.*

The assumption on the aerosol form has been added to the text.

*Sect. 1, 3rd paragraph:*
*The second sentence ("Known existing ... (Damadeo et al., 2013)" is awkwardly worded.*

The text has been revised in accordance with the reviewer's comment.

*Sect. 2.1, 1st paragraph:*
*The solar/lunar occultation mode of the SCIAMACHY instrument is mentioned here – are those observations usable for this study? If the measurements have sufficient quality, they would certainly add to the limited number of coincident occultation + limb*

*scattering measurements.*

SCIAMACHY solar and lunar occultation measurements were done around 60° latitude in both hemispheres. Since currently the PSD product is limited only by the tropical region (20° N –20° S), there is no possibility to include those in the study. Additionally, yet there is no PSD product from SCIAMACHY occultation measurements. However, in the future the synergistic use of the both modes will be tested.

*Sect. 3.1, 1st paragraph:*
*As discussed earlier, the failure to consider the consequences of any PSD other than single mode log-normal is a significant limitation of this study (just as it is a significant limitation of most limb scattering aerosol retrieval work to date). Even a single example testing this approach for another type of PSD (realistic, but not single mode log-normal) would add significant value to this study, by providing some indication of how much the results presented depend on that restriction.*

We chose to use uni-modal log-normal distribution because that is the one, which is assumed for our PSD retrieval algorithm. Additionally, uni-modal log-normal distribution was assumed for such instruments as SAGE II and OSIRIS. The suggested by the reviewer study is interesting as itself, but is outside of the scope of this paper.

*Sect. 3.2, 5th paragraph:*
*It would be helpful to provide some guidance about the sigma value that corresponds to w = 0.01 microns for a few examples, since this is an unconventional way to describe the PSD.*

The $\sigma$ values corresponding to some combinations of $R_{mod}$ and $w$ were provided.

Additionally, in Sect. 3.1 the formula of $w$ calculation is presented (see Eq. (2)).

*Sect. 3.2, 7th paragraph:*
*The basis for the perturbation analysis presented is confusing: Apparently the temperature and pressure are treated as variables that can be perturbed independently (without regard for hydrostatic balance, for example)?*

Although the reviewer is right, that the temperature and pressure are not independent, for this particular study it does not play a major role. For SCIAMACHY PSD product, ECMWF operational analysis data for the specific date, time and location of each SCIAMACHY limb measurement were used. The same data was used in this study. This product has about 10% uncertainty for each of the variables. The other possible datasets (e.g. MERRA) have a similar uncertainty. Thus, it was decided to change pressure and temperature independently. This explanation was added in the manuscript.

*Figure 9:*
*This figure illustrates some limitations on the conclusions that can be drawn from this analysis due to the constraints that have been imposed on the PSD during this analysis: What does it mean to perform a retrieval based on a single PSD, then analyze the variation of Angstrom coefficient with altitude (which should be zero, if a single PSD truly characterizes the stratospheric aerosol)? Or am I misinterpreting something in the methodology?*

Because of the poor wording in the original manuscript, the reviewer misinterpreted the methodology of the study. The PSD parameters were first retrieved and then $Ext$ and Ångström exponent was recalculated from it. Since the PSD varies, Ångström

exponent varies as well. We added more clear explanations to the text regarding the way Ångstöm exponents were obtained.

*References:*
*Yue, G. K.: A new approach to retrieval of aerosol size distributions and integral properties from SAGE II aerosol extinction spectra, J. Geophys. Res., 104, 27 491–27 506, 1999.*

References:
Malinina, E., Rozanov, A., Rozanov, V., Liebing, P., Bovensmann, H., and Burrows, J. P.: Aerosol particle size distribution in the stratosphere retrieved from SCIAMACHY limb measurements, Atmos. Meas. Tech., 11, 2085-2100, https://doi.org/10.5194/amt-11-2085-2018, 2018.

---

## Author Comment (AC2) · 31 May 2019

We thank the reviewers for the time they spent thoroughly reading the manuscript and commenting on the paper. We hope we have answered the reviewers' questions and improved the explanations where needed. To distinguish the referees' comments from the author's responses, the comments are shown in italicized font and the responses are highlighted in blue.

**General comments:**
*This paper reports the continuation of a previous work published by Malinina et al. (2018) presenting a new dataset of particle size information retrieved from SCIA-*

*MACHY. It proposes an extension of the dataset including the Angström coefficient and the aerosol extinction coefficient in the tropical zone (20°S - 20°N), as well as a sensitivity study and error analysis of their dataset. Afterward, the authors explore the link between the Angström coefficient and the parameters characterizing the particle size distribution (PSD). Based the sensitivity study, the authors claim that limb viewing instruments are more accurate for PSD retrieval than occultation instruments.*

*If the approach chosen by the authors provides an interesting and valuable insight into the problem of aerosol size retrieval from spaceborne instruments, I cannot agree at all with their conclusions on the sensitivity study. This conclusion is based on a too limited and biased analysis, which doesn't take (sufficiently) into account critical elements such as the impact of the bias induced by the assumptions made on the PSD in the forward model, and the influence of the covered spectral range and of the use of multiple wavelengths on the information content.*

*Further, the authors propose a long discussion about the usability of the Angström coefficient to derive size information and about the ill-posedness of the problem. Beyond long developments that are in some cases not really new or relevant, many arguments used in this discussion are truncated or even wrong, and this study should benefit a lot from a more accurate reading of the literature on this subject.*

*If the authors want to compare the capabilities of occultation and limb viewing experiments, they need to revise thoroughly their sensitivity study to take into account all aspects of the inversion problem, including what was published in the past on this topic. As presented here, most of the conclusions of the authors, including the conclusion that limb-viewing instruments are more accurate than occultation instruments cannot be drawn, and are thus basically wrong.*

After the thorough reading of the reviewer's comments we realized that there was a major misunderstanding of the PSD retrieval algorithms from the limb measurements. If we understand the reviewer correctly, she/he is talking about the PSD approximation in the forward model for the aerosol extinction retrievals. However, for the PSD

retrievals this assumptions are irrelevant, since there is no such intermediate step as extinction retrieval. After re-reading the manuscript, we realized that the text was misleading, so it was corrected accordingly. The misunderstanding on the algorithm also led to the misunderstanding of the Ångström exponent discussion and the sensitivity studies. With respect to the last one we agree, that the chosen wavelength interval was too short. In the revised manuscript we included shorter wavelengths. More details can be found in the "Detailed comments" section.

***Detailed comments:***
*Abstract:*

- *L. 5-6, p.1, "These uncertainties can be mitigated...": I am not sure that the mitigation is very efficient, because the assumptions made on the PSD for the forward model obviously precede the PSD retrieval, thus influences the PSD retrieval.*

While working on the reviewer's comments we realized that reviewer seems to misunderstand the retrieval algorithms used by limb scatter instruments. For limb scatter instruments the radiances are used directly to retrieve PSD products and, thus, unlike occultation instruments, do not require extinction retrieval as an intermediate state. Thereby, the uncertainties are mitigated if PSD information is retrieved.

- *L. 6-8, p.1: I don't agree with this statement. See discussion later on the body of the manuscript.*

Following the discussion in the body of the text the sentence has been revised.

- *L. 12-14, p.1: This is not a correct and complete estimate of the error on the extinction and Angstrom coefficient. This statement has to be revised. See comments on Section 5.1.*

Please, refer to this section for the detailed explanation. Here, we replaced the term "error" by "parameter error" to be more precise.

- *L. 16-17, p.1: Since OSIRIS is based on the same measuring technique (limb viewing geometry) and makes use of very similar assumptions on the PSD as SCIAMACHY in the forward model, the results of this comparison have to be considered very cautiously. This should be mentioned by the users. See also remark on Section 5.3.*

Please, refer to the discussion to this remark.

- *L. 19-21, p.1: Also, the Angström coefficient depends on the considered spectral range. This should also be mentioned.*

The sentence has been revised. We added the information that our studies are valid for an Ångström exponent for a single wavelength pair.

*1. Introduction:*

- *L. 4-5, p.2: I don't understand this sentence: the role of stratospheric aerosols on what ? There is an abundant literature about stratospheric aerosols, and stratospheric aerosols are the scope of a SPARC (Stratospheric Processes and their*

*Role in Climate) activity called SSiRC (Stratospheric Sulfur and their Role in Climate, See Kremser et al., 2016). The authors mention many works addressing the role of aerosols in climate, in the specific case of Asian summer monsoon, in geoengineering, and many other aspects. The aerosol evolution and role in the radiative forcing of the atmosphere was also the subject of several publications(e.g. Bingen et al., Remote Sens. Env., 2017; Brühl et al., Atm. Phys. Chem., 2018). On the other hand, it is true that the current period is characterized by a particularly low amount of satellite experiments with profiling capabilities. If this is what the authors want to (rightly) emphasize, they should reword their sentence. Otherwise, they should remove this sentence.*

The sentence has been revised.

- *L. 18, p.3: GOMOS processed with the AerGOM algorithm provides the extinction coefficient in the range 300-750 nm (Vanhellemont, et al., 2016, op. cit.).*

The information has been added in the text.

- *L. 18-20, p.3: The conversion of the basckatter coefficient to extinction is not a straightforward process since it requires the knowledge of the lidar ratio, a time-, space- and aerosol composition-dependent parameter. Errors on this parameter can thus induced a large variability and a large uncertainty on the derived extinction. This fact should be mentioned in the present discussion. Several publications address this problem, e.g. Rogers et al., Atm. Meas. Tech, 7, 4317-4340, 2014.*

The uncertainties in $Ext$ from CALIOP have been mentioned in the revised manuscript.

*2. Instruments and data:*

- *L. 15-16, p.4: how was the fixed particle number density determined from EC-STRA ? ECSTRA is a climatology of stratospheric extinction based on a parameter describing the overall volcanic state of the atmosphere, but does not provide size parameters. Please describe your methodology.*

The original text in the manuscript was inaccurate. $N$ was chosen in accordance with background ECSTRA climatology. Thus, assuming unimodal lognormal distribution with $r_{med}$=0.11 $\mu$m and $\sigma$=1.37 $N$ was recalculated from ECSTRA $Ext$ values. The text of the manuscript was changed accordingly.

- *L. 20, p.5: It should be mentioned that the approach proposed by Thomason et al, 2008 concerns the non-volcanic case.*

In the paper by Thomason et al. (2008) multiple ways of surface area density calculations are presented. They include operational formula, as well as two methods for non-volcanic cases. For this reason text of our manuscript has not been changed.

*3. Sensitivity of measurements to aerosol parameters*

- *L. 26-27, p.5: "unimodal" and "lognormal" are two independent concepts. The authors should add something like: ", here with a lognormal function:"*

The text of the paragraph has been changed in accordance with the comments of both reviewers.

- *L. 28, p.5: It might be useful to mention units, especially for the particle number density.*

We disagree with the reviewer on this point. Since $r$ and $r_{med}$ can be presented in any units of length (L), $N$, which has a dimension of L$^{-3}$, can represented by any units of length as well. The range of SCIAMACHY $N$ values, which was used in the study, is presented in Sect. 4.1.

- *L. 6-7, p.6: I don't understand this statement. Several degrees of freedom are required to retrieve the extinction coefficient at several wavelengths. It may look like extinction retrieval requires less degree of freedom than PSD retrieval, but, as explained at the end of the section, this is due to the fact that a very significant information content is hidden in the model used, more particularly in terms of PSD assumed in the forward model, including a distribution function and the related mode parameters. Consequently, the authors should qualify this statement, make the link with the important reminder at the end of the section, and at least precise if they are only considering the case of limb viewing instruments for which additional information on the PSD is provided in the forward model.*

We agree with reviewer that for the retrieval of $Ext$ at multiple wavelengths, several independent pieces of information are needed. However, in that particular sentence of the manuscript we were talking about $Ext$ coefficient at one wavelength. The sentence is there just to introduce the Eq. (3) and the link between the PSD and $Ext$. We have changed the sentence to make it more clear.

- *L. 8-9, p.6: The formulation of the extinction coefficient is not correct. It has to be written as the integration of the PSD weighted by the extinction cross-section (See, for instance, d'Almeida et al., Atmospheric Aerosols, Deepak Publishing, 1991). Eq. (3) amounts to considering that the extinction cross-section can be approximated by its value for a fixed particle radius $r_{med}$ and a wavelength $\lambda$, and put out of the integral, what in general is not correct. The extinction cross-section is a parameter describing the extinction of light with wavelength $\lambda$ by a single particle characterized by its radius and index of refraction. Hence, it doesn't depend on aerosol mode parameters. It is very important to clarify how $\beta_{aer}$ is computed because it determines how to interpret the error assessment for the extinction in §4.1.*

We disagree with the reviewer in that matter. The formulation of $Ext$ is correct, in case the unimodal lognormal distribution is assumed and Mie scattering theory is used. In Mie theory, which is used for the aerosol parametrisations, the cross-sections are calculated using the integration over the given PSD.

It is true, that the usual aerosol extinction formulation is

$$Ext_\lambda = \int_0^\infty Q(r,\lambda)\pi r^2 \frac{dN}{dr}dr = \int_0^\infty \hat{\beta}_{aer}(\lambda)\frac{dN}{dr}dr,$$

where $Q$ is the aerosol extinction efficiency, $N$ particle number density and $\hat{\beta}_{aer}$ is aerosol extinction cross-section for the particle with radius $r$. However, when the integration is done over the whole particle size range with the assumption of unimodal lognormal distribution, the above described expression is transferred to

$$Ext_\lambda = \beta_{aer}(r_{med}, \sigma, \lambda)N.$$

The refractive indices for our research were calculated using OPAC database (Hess et al., 1998), which was explicitly mentioned in the cited Malinina et al. (2018) and Rieger

et al. (2018) papers. However, we added the information on the assumptions on $Ext$ calculations to the sentence marked by the reviewer.

- *L. 6-8, p.7: This is confusing. Are the authors talking about the occultation case (Eq. (5))?*

As it was written in the original text we are talking about $I_{dir}$, which is an occultation case. We tried to make the sentence more clear.

- *L. 11-13, p.7: Since the solar scattering angle influences the phase function which depends critically on the PSD (the phase function being a weighted integration of the PSD), this issue has to be carefully investigated.*

The issue has been carefully investigated in the cited papers. The text has been changed accordingly.

- *Eq. (7) and (8): The parameter K is different in both equations and has actually different dimensions in both cases. Hence, a different notation should be used, for instance, $K_{obs}$ and $K_{scat}$.*

The parameters have been changed to $K_{dir}$ and $K_{dif}$.

- *L. 23, p.7: There is a factor $\pi$ missing in the expression of $K$, related to the particle cross-section $\pi r^2$.*
According to information on the page 19 (end of the first paragraph) in the cited Twomey (1977) book, the formula is correct.

- *L. 26, p.7: "Showed it": What did they showed?*

In that sentence we meant that Thomason and Poole (1993) showed K. We revised the sentence to make it less confusing.

- *L. 25-27, p.7: I am not sure that these details are useful: it seems that Thomason's formulation is different although it is the same, admittedly based on another choice of variable (volume of aerosol per volume of air, instead of particle radius). The authors might consider just mentioning that Thomason and Poole use a similar formulation instead of emphasizing the differences, in order to avoid confusion.*

In our opinion it is important to highlight that our sensitivity studies as well as kernel assessment by Rieger et al., (2014) are not directly comparable to the study of Thomason and Poole (1993).

- *L. 1, p.8: There seems to be an error in $K$'s dimensions in the case of OSIRIS. If Eq. (7) is applied, $K$'s units should be expressed in W (with I in W/m$^2$ and n(r) in number of particles per volume unit). Even if applying the expression derived by Rieger et al. (2014), a radiance factor appears in the expression. Consequently, $K$ should include the contribution of the energy flux, and not be dimensionless.*

[Figure]

There is no error (please, see Eq. (7) and its derivation in Appendix A in Rieger et al., (2014)). Rieger et al. (2014) were not working with the radiance, but with measurement vector defined as $y = ln(\frac{I_{aer}+I_{Ray}}{I_{Ray}})$. That resulted in the dimensionless $K$.

- *L.14, p7 -L. 11, p.8: Overall, this discussion is a bit confusing: It seems that the aim is to show that in all cases, the inversion problem can be formulated using a similar expression, either using Eq. (7) for the occultation case, or Eq. (8) for the limb case. But immediately afterward, it is explained why this model is considered as much too simple in the OSIRIS case, and how it is not suited and will not be used for sensitivity studies in the case of SCIAMACHY. What is then the utility of this discussion?*

The aim is to show which studies have been done before and why we do not follow this particular way. Additionally, it should be noted, that $K$ is not related to the inversion problem formulation.

- *Eq. (9), p.8: This separation between "aerosol" and "Rayleigh" signals supposes that there is a clear distinction between both, and the authors most probably infer this Rayleigh signal from ancillary data of air density, temperature and pressure. The reality is much more complex, since very thin aerosol particles are also Rayleigh particles and their contribution to scattering cannot be discriminated from the molecular Rayleigh compound. This is especially the cases for "thin aerosol" cases considered by the authors. The only way to separate both contributions is to rely on the meteorological data that might be inaccurate with respect to the local condition encountered by the spaceborne instruments.*

[Figure]

In our study we can distinguish between the aerosol and Rayleigh signal, because we use the model data. This fact is explicitly highlighted two sentences later. The reviewer is absolutely right assuming, that we calculate the Rayleigh signal from ancillary meteorological data. Since the meteorological data contains certain uncertainties, we introduce the "low sensitivity area" and justify it's threshold while discussing the results.

- *L. 5-11, p.8: This model only takes into account the Rayleigh and aerosol compounds. How do the other contribution (trace gases) interfere in this analysis?*

For the original study, where only 750 nm and 1530 nm were taken into consideration, the influence of the trace gases was negligible, because those wavelengths are outside of absorption lines. For the new study which includes 386 and 525 nm sensitivity, ozone climatology was taken into consideration. This information is included into the revised manuscript.

- *L. 12-13, p.8: After the previous discussion, we know that Eq. (8) will not be used, but not which model/expression/equation was actually used to quantify the sensitivity. This should be clarified.*

We could not follow what reviewer meant in this particular comment. The formula for $S$ was introduced 3 lines above. Additionally, in the highlighted sentence we name and provide the reference to the model we used in our studies. To make it more clear we deleted the paragraph separation before this sentence.

- *L. 17, p.8 -L. 14, p.9: This sensitivity study is limited to the sensitivity of individual extinction channels, and this for two wavelengths in the infrared, including the*

*1530 nm-wavelengths, which is much higher than the particle size range the authors consider as relevant (50 - 300 nm). The behaviour of the sensitivity curve S and its quantitative assessment is absolutely insufficient to assess the performances of (existing) limb viewing instruments versus occultation instruments, for several reasons:*

– *If it is true that the size distribution influence twice the expression of the radiance (Eq. 6, through the scattering coefficient and the phase function), the error made by assigning inaccurate values of the aerosol mode parameters affects equally twice this parameter.*

– *The value of the wavelength influences significantly the scattering efficiency, as illustrated on the figure above. In particular, if the wavelength is very large compared to the particle radius, aerosol particles behave as Rayleigh particles, scattering becomes independent of the particle size (scattering $\lambda^{-4}$) and the size parameter cannot be discriminated. The figure shows the dependence of the scattering efficiency as a function of the parameter $x=2\pi m/\lambda$ (see e.g. van de Hulst, Light Scattering by Small Particles, Dover Publications, 1957), where the refraction index m is representative for a 75% $H_2SO_4$-25% $H_2O$ aerosol composition and the spectral range 350-1530 nm. It also shows,for particle size range 50-300 nm used on Figure 1 of the paper, the range of x parameter covered for the two wavelength considered by the authors ($\lambda$= 750 and 1530 nm, in red and magenta) and two other wavelengths representative for the spectral range covered by the most occultation instruments observing in the UV-visible-near IR range ($\lambda$= 350 and 500 nm, in cyan and green). It is obvious that the dynamic range corresponding to 1530 nm is particularly reduced (and similar to the Rayleigh regime). The 750 nm channel provides more variability in the scattering efficiency curve than the one at 1530 nm, but with a large overlap with the dynamic range of this first channel. On the other hand, wavelengths of 500 nm and 350 nm*

*provide a much larger dynamic range covering almost all possible values of the scattering efficiency between 0 and more than 4 (van de Hulst, op. cit., 1957).*

– *The authors don't take at all the critical aspect that particle size retrieval can be retrieved by the combination of extinction values at several values. In this respect, it is clear from the figure above that the dynamic range covered by the wavelengths spread over the UV-visible-near IR range like for instruments such as SAGE II and III, POAM III, or GOMOS provides a much larger information content than the combination 750-1530 nm. This aspect is of crucial importance in the case of real aerosol particle population where the diversity of particle sizes blurs the scattering response of individual particles, especially when several aerosol modes are present simultaneously (i.e. with a large mode width of the equivalent lognormal size distribution). The larger the spectral range covered in the Mie regime, the higher the information content.*

*As a conclusion of this discussion, even if the calculation of the modelled sensitivity is correct, it is not sufficient to assess the performances of a technique and it doesn't provide definitive arguments to conclude about the comparison of the overall capabilities of (existing) limb viewing versus occultation instruments.*

If we understand the comment of the reviewer correctly, there was a misunderstanding about the algorithm. As we specifically mentioned in Malinina et al.,(2018) we do not use aerosol extinction coefficient to retrieve PSD parameters. Instead, as we wrote in Sect. 2.1, limb radiances were used directly to obtain $R_{mod}$ and $\sigma$. Although the logarithms of the occultation radiances are proportional to aerosol extinction coefficient, for limb radiances the dependency is more complex. This fact was additionally highlighted in the Sect. 3.1 and shown with the Eqs. (5) and (6). Thus, reviewer's comment on the increased error by the wrong assignment of the aerosol mode is irrelevant for the limb radiances.

We agree with the reviewer that the spectral range presented in the original manuscript might be incomplete for the conclusions we've made. Thus, to make the study deeper we added in the revised manuscript the sensitivity assessment at $\lambda$=386 nm and $\lambda$=525 nm. These particular wavelengths were chosen because they were employed by SAGE II instrument. Summary of our new study is following: $\lambda$=525 nm does not provide much of additional information about the smaller particles for the occultation measurements, but $\lambda$=386 nm does. Thus, in order to obtain information on smaller particles by the occultation instruments the wavelengths about 386 nm should be taken into consideration. For more details, please refer to the revised manuscript.

The reviewer assumed that we did not consider the combination of the multiple wavelengths (radiances or extinctions) to retrieve the PSD information. However, even if a combination of the wavelengths is used, e.g. to obtain $r_{eff}$ by SAGE II ($\lambda$=525 nm and $\lambda$=1020 nm), it does not increase the sensitivity to the small particles as both of the wavelengths are insensitive to them.

At the end, we want to highlight, that this study was not performed to compare the overall capabilities of the limb and occultation instruments, but to show with a different approach the 0.1 $\mu$m sensitivity limit presented by Thomason and Pool (1993) and compare this sensitivity to the ones from limb instruments. For this purpose model studies are sufficient.

- *L. 8-9, p.9: 0.1 $\mu$m is not a magic limit for the sensitivity of occultation instruments, but it corresponds to some upper limit of the Rayleigh scattering regime. It depends thus on the spectral range covered by the instrument.*

We agree with the reviewer that 0.1 $\mu$m is not a magic number. It is a result of the use of the combination of the radiances at $\lambda$=525 nm and $\lambda$=1020 nm by SAGE II. But it should be highlighted that for now SAGE II is the only occultation instrument

providing the PSD information in the peer-reviewed publications. Thus, we do not see any contradictions.

- *Figure 2, p.9: Both figures compare the limb and occultation geometries in a spectral range limited to wavelengths values where the occultation geometry is particularly insensitive, and that doesn't correspond to the spectral range covered by most occultation instruments (See above). It is actually visible that the sensitivity in the occultation case is increasing toward the smallest wavelengths. Consequently, this comparison is biased and it doesn't reflect the true sensitivity of real sensors used for aerosol remote sounding from space.*

We agree with the reviewer that the Fig. 2 was relatively biased with respect to the occultation measurements. As we wrote in the answer to the above mentioned comment, we added $\lambda$=386 nm and $\lambda$=525 nm, the wavelengths used by SAGE II.

- *L. 19-25, p.10: Taking into account the various elements cited above, I cannot agree at all with these conclusions.*

We revised the conclusions based on the analysis of the sensitivity of the radiances at $\lambda$=386 nm and $\lambda$=525 nm.

*4. Error assessment*

- *L. 15-17, p.11: The chosen scenarios cover quite nicely aerosol particle populations with radii up to about 250-300 nm. One should still bear in mind that volcanic aerosols, in view of existing estimates from satellite and balloon-borne*
*datasets, may reach significantly higher values during particular (volcanic) periods the SCIAMACHY lifetime.*

The only known to us datasets, which were published in the peer-reviewed journals, cover the period from 2002 to 2012 and provide PSD information, are the ones from SAGE II and OPCs. The official SAGE II provides effective radius $r_{eff}$, which does not have unique translation into $r_{med}$ or $R_{mod}$ and, thus, is not easy to interpret the particle sizes based on that value. The SAGE II dataset published in Bingen et al. (2004) covers the period before 2002. OPCs provide $r_{med}$, $\sigma$ and $N$, however, the PSD is considered to be bimodal lognormal. Indeed, $r_{med}$ for the coarse mode can increase 0.25 $\mu$m. Though, in our estimation PSD is considered to be unimodal lognormal, and in that case $r_{med}$ is somewhere in between the fine and coarse mode fitted by OPCs. Thus, the mode radii of about 0.20 $\mu$m seem to be reasonable for the considered period. If the reviewer knows any other published PSD datasets, which are covering the period from 2002 to 2012, we kindly ask to provide the reference.

- *L. 15-22, p.11: If I understand well, the errors (calculated as the "median relative error") reflects the variability obtained from an ensemble of simulations where Gaussian noise was added to the synthetic radiance. On the other hand, the synthetic scenarios are constructed using a fixed size distribution (i.e. a fixed choice of $r_{med}$ and $\sigma$) and only N is supposed to decrease exponentially as described in L. 9, p.11. The realistic character of this profile is thus very relative. Consequently, the quantity investigated here is very different from the extinction uncertainty, that include experimental, instrumental, modelling and other retrieval errors into account. The use of the term "error" is thus particularly misleading, and should be replaced by something more adapted.*

The reviewer understood our approach to the error assessment correctly. However, in

the manuscript we assess so called parameter error (see Rodgers (2000), von Clarman et al., in preparation) rather than the full uncertainty. We revised the text to make this term more clear.

- *L. 1-4, p.12: See remark on L. 8-9, p6.*

Please, see the answer to L. 8-9, p6.

- *Eq. (10), p.13: See remark on L. 8-9, p6.*

Please, see the answer to L. 8-9, p6.

- *L. 22, p.13-end of p.14: Same remark as for L. 15-22, p.11.*

See the answer to the comment L. 15-22, p.11. We revised the text to make the term "parameter error" more clear.

*5. Comparison of the measurements results*

- *L. 10-11, p.15: The difference in the measurement technique is not an issue at all! It is the essence of validation efforts to use different datasets, including datasets based on different measurements techniques, to assess the strengths and weaknesses of the measurements to be investigated. This statement is thus inappropriate and should be removed. Arguments developed here may be pertinent to discuss the origin of possible weaknesses and strengths, but not as some*

[Figure]

*kind of a priori disclaimer to relativise the adequateness or validity of a comparison exercise.*

We disagree with the reviewer in that matter. The difference in the measurement technique is an issue. This issue is related to the difference in the radiative transfer and as a result different sensitivity of SCIAMACHY and SAGE II to the small particles. Thus, it was known for the comparison presented in Malinina et al. (2018) that there will be differences between $r_{eff}$. Here, we tried to present more appropriate comparison between the instruments. We believe, that presenting comparison of extinction coefficients from SAGE II (accurate value) and aerosol extinction coefficients recalculated from SCIAMACHY PSD product is more suitable than the $r_{eff}$ comparison. We did not try to provide the reader the impression that the comparison is irrelevant, in turn we tried to do it scientifically more correct based on the information we have.

- *L. 12-13, p. 15: As explained above, the sensitivity analysis proposed here is biased and insufficient, mainly because it doesn't take into account the whole spectral range taken covered by the occultation instrument, here SAGE II. Hence, even if it has been shown, indeed, that SAGE II is less sensitive to thin particles than to particles in the range 0.25-0.40 μm (See for instance Bingen et al., Ann. Geophys., 2003 for a comparison with balloonborne measurements) so that $r_{eff}$ is indeed expected to be biased high in the case of SAGE II, a more rigourous analysis taking into account all aspects of measurements and retrieval (including the impact of the different assumptions and approximations made in the limb viewing case) is needed to draw definitive conclusions about comparisons between SCIAMACHY and SAGE II.*

We got the impression, that the reviewer misunderstood the PSD retrieval algorithm from SCIAMACHY. Again, we do not use the aerosol extinction coefficients, but the

radiances. Thus assumptions on the PSD in the limb viewing case are irrelevant for that studies. Further assumptions were already addressed in Malinina et al. (2018). With respect to the sensitivity, please refer to the answers to Sect. 3 and the revised Sect. 3.

- *L. 28, p.15: "the standard error on the mean relative difference" might be more clear.*

The reviewer might have misread the sentence. In the manuscript it is "the standard error OF the mean.", which is a statistical term.

- *L.28-29, p.15: Both parameters provide different information, and the one presented should be the most appropriate to assess the quality of the agreement between datasets ! If the standard deviation is so large that it makes the figure very busy, it means that the quality of the profile-to-profile comparison is very poor, and this should also be shown in some way !*

Standard error of the mean is calculated as standard deviation divided by square root from the amount of observations. In the beginning of the the section we provide the amount of collocations, so the standard deviation can easily be recalculated if a reader needs this particular value. We do not show standard deviation, because at the altitude range from 24 to 31 km the relative differences are quite close to each other and the lines even cross sometimes. Thus, plotting standard deviation will make the plot unreadable.

- *L. 33-34, p.15: I don't see why a similar behaviour is expected for the mean relative difference of the extinction at 1020 nm and $r_{eff}$: the extinction at 525 nm, which shows a quite different behaviour on Figure 5, plays an equally important role in the computation of SAGE II's effective radius (Thomason et al., 2008, op. cit.).*

In the paper by Thomason et al. (2008) there is no formula for $r_{eff}$ derivation provided, it is just noticed that $Ext_{1020}$ and $Ext_{525}$ are used to derive PSD information. In the cited in Thomason et al. (2008) paper by Chu et al. (1989) there is also no formula given. Additionally it should be noted that for surface area density it was shown that $Ext_{1020}$ plays bigger role than $Ext_{525}$. Thus, we are not sure that $Ext_{525}$ is equally important. However, even if it is the case, we expect $r_{eff}$ to show similar behavior to the components it is calculated from, except they are not absolutely anticorrelated. And this is not the case for $Ext_{525}$. For that reason, we do not think that the similarities in $r_{eff}$ and $Ext_{1020}$ is something unexpected.

- *L. 34, p.15: What is the usefulness of the reference to Sect. 2.3 ? There is nothing more about $r_{eff}$ is Sect. 2.3 than what is said here. A reference to some paper where the derivation of $r_{eff}$ is presented would be more useful.*

In the answer to the previous comment we noted, that we could not find the exact formula for $r_{eff}$. In Sect. 2.3 we cite Damadeo et al. (2013) and Thomason et al. (2008), who notice that both, $Ext_{1020}$ and $Ext_{525}$ are used. If the reviewer knows any other paper or source of the information on how $r_{eff}$ is calculated, we will be happy to include it into the next revision of the manuscript.

- *L. 4-10, p.16: This figure only shows that there is a bias of about 8% between*

*the two ways used to derive the extinction coefficient from the PSD. This is very different from providing any assessment of the error on the extinction coefficient. Further, the uncertainty on the PSD has, to my knowledge, not been correctly characterized: available "errors" (Malinina et al., op. cit., 2018) are derived in a similar way as the extinction "error" in the present paper and similarly express a variability with respect of an ensemble of more or less realistic synthetic cases (See remark on L. 15-22, p.11). Hence, I am not very sure Figure 6 adds more information on the extinction uncertainty, other than emphasizing some incoherence in the processing chain producing Ext, the PSD and the Angström exponent, most probably related to the successive assumptions and approximations made.*

In this passage we haven't tried to assess errors. We exactly wanted to show the differences in recalculation of $Ext_{750}$ with Ångström exponent in comparison to the "true" value. When comparing the measurements from different instruments providing aerosol extinction at different wavelengths, Ångström exponent is used to recalculate the extinction coefficient (e.g. von Savigny, 2015; Rieger et al., 2018). Thus, we think it is important to show numerically the incertanies in this recalculation. With respect to errors, as we noted in the above mentioned comment, we assessed the parameter error. The same as was done in Malinina et al. (2018).

- *L. 16, p.16: What do the authors mean ? The green curve corresponding to the $Ext_{1020}$ in Figure 5 and the blue curve corresponding to $a_{525/1020}$ have completely different shapes !*

We meant that $\alpha_{525/1020}$ is almost mirrored from $Ext_{1020}$. However we agree with the reviewer, that the phrasing was not correct and we revised the section.

[Figure]

- *L. 2, p. 17: What do the authors mean by "the bias in this comparison" ? I guess they just mean something like "this behaviour" ?*

The sentence has been changed to "The differences observed in this comparison are expected".

- *L. 5-7, p. 17: I am not sure that adding complexity to the problem by extending the latitudinal range will help answering this question. A better way is probably to carefully examine the impact of every assumption and approximation made in the SCIAMACHY retrieval, and its possible altitude dependent character. Using real and totally independent measurements like balloonborne OPC measurement profiles to reconstruct a forward model and every step of the retrieval might help understanding the remaining issues.*

We agree with the reviewer that comparison of SCIAMACHY dataset with OPCs will be beneficial. However, the longest series of OPCs measurements is presented in Laramie, which is located in the mid-latitudes. During the SCIAMACHY lifetime there were 3 tropical OPC campaigns, however to provide an extensive comparison the Laramie dataset should be taken into consideration. Thus, to compare SCIAMACHY and OPC profiles, the existing algorithm should be applied to the mid-latitudes. Additionally, that will increase the amount of collocations with SAGE, which will allow to provide statistically significant comparisons.

- *L. 15, p.19: I don't agree with that. As explained above, 8% is the difference that was used between the two methods used here, but it does not represent the uncertainty on the recalculated $Ext_{750}$. What can be concluded from this*

*calculation is that the additional uncertainty on Ext$_{750}$ due to the recalculation is at least 8%.*

Please, refer to the answer to the L. 4-10 p.16.

- *L. 1-4, p.20: It is very likely that, beyond the same measurement technique and the same use of spectral information, the similarities between the retrieval schemes, including the assumptions and approximations made, greatly contribute to the similar performances on both datasets. Hence, both SCIAMACHY and OSIRIS datasets might present similar biases, which are impossible to discern from the comparison SCIAMACHY-OSIRIS but potentially (very) significant. In this way, the comparison might be (very) "unfair" with respect to SAGE II and lead to the possible wrong conclusion that SCIAMACHY and OSIRIS are likely to be more accurate than SAGE II. It is very important to add such a discussion point here to exclude wrong conclusions on the respective degree of reliability of SCIAMACHY, OSIRIS, and SAGE II.*

Yes, both SCIAMACHY and OSIRIS use the same measurement techniques and the same wavelength information. However, the algorithms are quite different. Firstly, OSIRIS fits $r_{med}$ and $N$ with the assumption of $\sigma$=1.6 from the measured radiances and from that the extinction coefficients and Ångström exponent are calculated (please, see Sect. 2.2). For SCIAMACHY, $N$ is assumed and $R_{mod}$ and $\sigma$ are fitted directly from the radiances and from those parameters aerosol extinction coefficients are calculated. So even the general way is the same (fix one of the PSD parameters $->$ get two others from the radiances $->$ recalculate extinction and Ångström exponent), the assumptions are absolutely different. Secondly, SCIAMACHY and OSIRIS have different geometries. Taking these differences into the consideration, we think that the observed differences in Ångström exponent are quite remarkable. It

should be also noted, that in the highlighted by the reviewer text we emphasize the remarkable SCIAMACHY-OSIRIS consistency because of the instruments similarities and do not try to reduce SAGE II reliability.

*6. Discussion*

- *L. 16, p.21-L. 3, p.22: I don't understand the rationale used here. We are dealing here with an underconstrained problem (as stated by the authors in L. 15-16, p.21) with 3 unknown: some radius and some spread characterizing the shape of the PSD, and a measure of the number of particles characterizing its amplitude. The way used to correctly constrain the problem was to fix the particle number for Malinina et al., and to fix the spread for Rieger et al. These are just two possible choices (amongst other possible ones), and combining the results of both approaches will not bring any additional information at all ! Further, using $R_{mod}$, w or $r_{med}$, $\sigma$ are just two equivalent ways to model the same thing, as nicely illustrated by the two panels in Figure 11.*

We agree with the reviewer that the presented ways to constrain the problem are not the only possible. This was presented in the manuscript in l. 2 p. 22. We also agree that combining the approaches will not add any additional information and the use of the pairs $R_{mod}$, w and $r_{med}$, $\sigma$ is the equivalent way to present the same Ångström exponent. However, in the manuscript it was not stated that combination of the methods will add information and it was highlighted that both panels of Fig. 11 present the same. That's why we do not know how to address this comment.

- *L. 1-17, p.23: If I understand well, the authors found that there are an infinite number of solution for a PSD giving a the same spectral dependence of the extinction as one value of the Angström coefficient. Since 3 parameters are used in*

*the PSD and only one for the Angström coefficient to describe the same kind of information (actually the spectral dependence of the measured signal), this result is quite trivial and I don't see the added value of such a long discussion. It would be much more interesting to explain how the authors intend to solve the problem in the case of SCIAMACHY. The way to deal with this problem was already addressed in the past (Echle et al., J. Geophys. Res., 103, 19993-19211, 1998; Fussen et al., Atmosph. Env., 35, 5067-5078, 2001), so that, if the authors are willing to present their own method applied to SCIAMACHY, they should at least refer to these works in the discussion.*

It seems to be that there is a misunderstanding. The purpose of the highlighted paragraph is to show that single value of Ångström exponent can not be used as a measure of the particle size. This was shown by presenting $\alpha_{750/1530}$ as a function of $R_{mod}$ and $w$ and $r_{med}$ and $\sigma$. Even though the result seems to be trivial and was briefly introduced by Ångström himself in (Ångström, 1929), this is often forgotten. In the scientific community, Ångström exponent at single wavelength is quite often used to discuss about changes in the particle size. It might not be as widespread in the literature, but is quite often used at conferences and at the discussion level. For that reason, we wanted to highlight this fact again and illustrate, why this belief is wrong. As we stated already, for the SCIAMACHY retrieval algorithm there is no problem of the extinction and Ångsröm exponent spectral dependency, because unlike for the occultation measurements, we retrieve PSD parameters directly from the limb radiances. This statement was explicitly highlighted in Malinina et al. (2018) and mentioned in section 2 of this manuscript. Thus, the suggested by the reviewer references are irrelevant for this particular paragraph. We added the particular purpose of the paragraph into the manuscript text.

- *L. 15-18, p.23: This part of the discussion are absolutely truncated. The authors are here finding that more than one Angström coefficient, or equivalently, more*

*than 2 extinction channels, should help deriving the PSD. There is absolutely nothing new in this, and telling that "all known space-borne instruments" provides only one Angström coefficient amounts to saying that all known space-borne instruments provides extinction at, at most, two wavelengths, what is obviously wrong. In particular, as mentioned above, extinction coefficients from occultation instruments are used over several wavelengths covering a large spectral range, what helps solving the ill-posedness of the problem (see e.g. Fussen et al., Atmosph. Env., 35, 5067-5078, 2001; Bingen et al., Ann. Geophys., 21, 797-804, 2002). Actually, this disappointed findings shows again that the comparison made here between occultation and limb viewing instruments is very incomplete, and the statement that limb instruments have a better potential or are more sensitive to the aerosol size is wrong or, at least, very premature.*

Here we meant, that usually in peer-reviewed publications and in the conference presentations the Ångström exponents at single wavelength are used directly to judge about the change in PSD, which is fundamentally wrong. As reviewer rightly notices, that can be done if multiple Ånstgröm exponents are provided. In particular that approach is helpful for the occultation instruments. However, to obtain the information on PSD from multiple Ångström exponents certain retrieval should be done. We have changed the formulations in the paragraph to make the purpose and the message more obvious.

- *L. 18-23 p.23: After an eruption, the PSD evolves according to the microphysical processes taking place, and using a Mie model, the evolving PSD determines unequivocably the extinction behaviour, and hence the Angström coefficient through Eq. (10). The inability to derive unambiguously the PSD from the Angström coefficient obviously doesn't implies the other way around. The authors seem to claim that the behaviour of the Angström coefficient is unpredictable and depends on*

*the detail of their formalism ("with $\sigma$ remaining unchanged"), but this is obviously
wrong and the point is that the exploration of Figure 11 just doesn't provide suf-
ficient information to foresee the behaviour of the Angström coefficient after an
eruption.*

The above marked paragraph is a summary of the whole section 6. In section 6 two
main topics were addressed. The first addressed topic was the tropical climatology
of $\alpha_{750/1530}$, which includes the discussion of the Ångström exponent changes after
the volcanic eruption based on Figs. 9 and 10. While analyzing those Figs. we have
noticed, that the distribution of $\alpha_{750/1530}$ after the volcanic eruption is very similar to
the one of $w$. That led to the discussion of the second topic, i.e., the dependence
of $\alpha_{750/1530}$ on the PSD parameters. Those topics are related, however they are
independent in their own way. In the manuscript we never suggested to use Fig. 11 to
predict the behaviour of the Ångstöm exponent after volcanic eruption. However, we
understand that the formulations used in that paragraph might mislead the reader. For
that reason, the paragraph was revised to resolve the issue.

*7. Conclusion*

- *Based on all what was explained above, I disagree on most of the conclusions
  given here. The reasons for this have been given before and don't require any
  repetition. I would just mention that the "most correct conclusion" in my opinion
  (L. 18- 19, p.24: "it is impossible to derive any reliable information (...)") is still
  incorrect: the information provided by the Angström coefficient is incomplete, but
  is reliable if the extinction measurements used for its calculation are reliable.*

We attempted to address all of the reviewer's comments. The conclusions for those
comments have been changed accordingly.

The phrase the reviewer calls here as incorrect was taken outside of the context. We

never meant that Ångström exponent provides unreliable information, we meant that "it is impossible to derive any reliable information on the changes in the aerosol size based solely on the Ångström exponent for one wavelength pair". The last was proven in Sect. 6.

***Technical corrections:***

• *L. 11, p.2: "periods of heavy aerosol loading" ?*

The marked by the reviewer phrase has been changed to "the periods with enhanced aerosol loading"

• *L. 34, p.2: Incorrect use of the parentheses; reference "Bingen et al" could be moved after "Known existing datasets" to make the text more fluent: "Known existing PSD data sets (Bingen et al. (2004)) were obtained (...) to 2005 (Yue et al., 1989; Thomason et al (2008) etc.).*

The sentence has been revised.

• *L.30, p.3: "errors in the extinctions" or "errors in Ext".*

The phrase has been revised.

• *L. 24, p.5: "following a lognormal distribution".*

Since there is neither grammatical, nor stylistic mistake in that phrase, we kept it unchanged.

- *L. 7, p.6: "by lack of information"?*

The suggested by the reviewer change will distort the meaning of the sentence. Because of that, it was left unchanged.

- *L. 6, p.7: Incorrect sentence.*

The sentence has been revised.

- *LL. 15, p.9: duplicated "the".*

The phrase has been revised.

- *L.3, p.10: "reasonable".*

We could not follow, what was wrong with the word "reasonable". The spelling was correct, so we kept the sentence unchanged.

- *Caption Figure 3: "The solid lines show the extinction calculated from PSD..." ?*
The sentence has been changed to "The solid lines show the $Ext_{750}$ profiles calculated from PSD product.".

- *L. 7, p.12: "latitudinal".*

The sentence is correct. We meant altitudinal.

- *L.8, p.13: "a subject for further studies" ?*

The phrase has been revised.

- *L. 28, p.15: The authors might add the colour used for $Ext_{750}(\alpha_{525/1020})$ to be complete.*

The color has been added.